# Functional reallocation of sensory processing resources caused by long-term neural adaptation to altered optics

**Antoine Barbot[1,2]\*, Woon Ju Park[3,4], Cherlyn J Ng[1,2], Ru-Yuan Zhang[3,5,6], Krystel R Huxlin[1,2,3,7], Duje Tadin[1,2,3,7], Geunyoung Yoon[1,2]\***

[1]Flaum Eye Institute, University of Rochester Medical Center, Rochester, United States; [2]Center for Visual Science, University of Rochester, Rochester, United States; [3]Brain and Cognitive Sciences, University of Rochester, Rochester, United States; [4]Department of Psychology, University of Washington, Seattle, United States; [5]Shanghai Key Laboratory of Psychotic Disorders, Shanghai Mental Health Center, Shanghai Jiao Tong University School of Medicine, Shanghai, China; [6]Institute of Psychology and Behavioral Science, Shanghai Jiao Tong University, Shanghai, China; [7]Department of Neuroscience, University of Rochester, Rochester, United States

**Abstract** The eye's optics are a major determinant of visual perception. Elucidating how long-term exposure to optical defects affects visual processing is key to understanding the capacity for, and limits of, sensory plasticity. Here, we show evidence of functional reallocation of sensory processing resources following long-term exposure to poor optical quality. Using adaptive optics to bypass all optical defects, we assessed visual processing in neurotypically-developed adults with healthy eyes and with keratoconus – a corneal disease causing severe optical aberrations. Under fully-corrected optical conditions, keratoconus patients showed altered contrast sensitivity, with impaired sensitivity for fine spatial details and better-than-typical sensitivity for coarse spatial details. Both gains and losses in sensitivity were more pronounced in patients experiencing poorer optical quality in their daily life and mediated by changes in signal enhancement mechanisms. These findings show that adult neural processing adapts to better match the changes in sensory inputs caused by long-term exposure to altered optics.

**\*For correspondence:**
antoine.barbot@nyu.edu (AB);
gyoon@ur.rochester.edu (GY)

**Competing interests:** The authors declare that no competing interests exist.

## Introduction

Understanding how we see requires insights into the contribution of both optical and neural factors mediating visual perception, from the processing of images formed on the retina to the resulting perceptual representations. Visual processing is fundamentally limited by the eye's optics, which determine retinal image quality and constrain performance. Human optics, however, are not fixed and can substantially change over our lifespan and in disease (*Artal, 2008*). Consequently, to achieve efficient perceptual processing, the brain must be able to continuously adjust to changes in sensory inputs over a range of different timescales and magnitudes. There is growing evidence that visual processing adapts to the presence of optical blur, altering neural processing to improve visual representations (e.g., *Artal et al., 2004*; *Sawides et al., 2011*; *Webster, 2015*; *Webster et al., 2002*). Neural adaptation mechanisms compensate for blur-induced reductions in physical contrast of high spatial frequency (SF) retinal signals, enhancing visual sensitivity for fine spatial details (*George and Rosenfield, 2004*; *Mon-Williams et al., 1998*; *Webster, 2015*; *Webster et al., 2002*). Even observers with typical (i.e., healthy) optical quality show evidence of neural adaptation to their

own optical blur, such that modest changes in their own optical aberration pattern severely degrade perceived image quality (*Artal et al., 2004*).

Distinct neural adaptation mechanisms operate over different timescales (*Bao and Engel, 2012*; *Bao et al., 2013*; *Glasser et al., 2011*; *Haak et al., 2014*). Most studies have investigated the effects of blur adaptation over short-term periods (i.e., minutes to hours), and often for low magnitudes of blur, thus limiting our understanding of how the adult human brain adapts to large changes in optical quality over longer periods of time (i.e., months to years). This, in turn, limits the development of clinical rehabilitations of a significant part of the population that chronically experiences abnormal optics in their daily life. A major experimental challenge comes from the difficulties of empirically isolating neural from optical factors. Technological advances in the field of adaptive optics (AO) offer a unique opportunity to bypass the limits imposed by optical factors, while directly assessing visual processing of images free from any optical imperfection (*Marcos et al., 2017*; *Roorda, 2011*). AO is a powerful technology that can be used to improve optical systems, including the human eye, by deforming a mirror to correct the wavefront distortions caused by the eye's optics. AO correction in 'healthy' eyes allows observers to detect a larger range of high SF information otherwise indiscernible in the presence of optical blur, improving contrast sensitivity (*Liang et al., 1997*) and visual acuity (VA) (*Yoon and Williams, 2002*). More importantly, AO correction can be used to assess how changes in optical quality alter neural functions by making it possible to assess visual processing under fully-corrected optical quality in both typical 'healthy' eyes (e.g., *Artal et al., 2004*; *Liang et al., 1997*; *Sawides et al., 2011*; *Yoon and Williams, 2002*; *Zheleznyak et al., 2016*) and those with severe optical abnormalities (e.g., *Sabesan et al., 2007*; *Sabesan et al., 2017*; *Sabesan and Yoon, 2009*).

In this context, keratoconus (KC) represents an ideal model of long-term neural adaptation to optical defects (*Figure 1*). KC is a severe corneal disease afflicting neurotypically-developed adults (*Vazirani and Basu, 2013*). The corneal stroma progressively thins and assumes a conical shape, resulting in a substantial increase in both lower-order (defocus and astigmatism) and higher-order

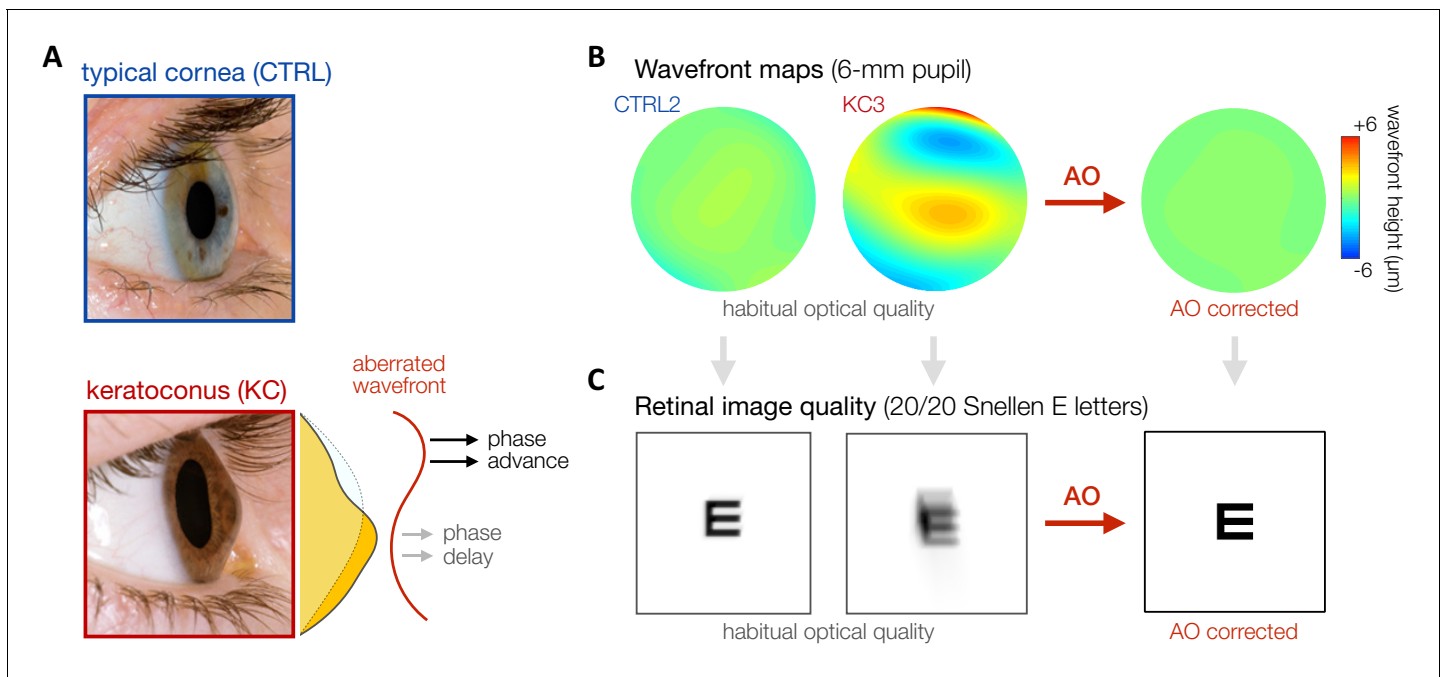

**Figure 1.** Keratoconus (KC). (A) KC is a progressive eye disease affecting neurotypically-developed adults, in which the typically round cornea thins and bulges into a cone-like shape. *Pictures adapted from* http://www.theeyefoundation.com. (B, C) As shown on the wavefront maps , KC results in large amounts of higher-order optical aberrations (HOAs) that cannot be fully corrected using conventional optical devices. As a result, KC patients are chronically exposed to large amounts of habitual optical aberrations and poor retinal image quality in their daily life (simulated for 20/20 Snellen E letter for 6-mm pupil). Adaptive optics (AO) techniques can be used to fully correct all optical aberrations during in-lab visual testing, allowing us to bypass optical factors and directly assess differences in neural visual processing.

optical aberrations (HOAs). KC is essentially a bilateral condition, although disease development may be highly asymmetric. The onset of KC occurs typically during the late teens or early twenties, with variable progression that can last until the third or fourth decade of life, when corneal shape generally becomes stable (*Vazirani and Basu, 2013*). Although HOAs are relatively small in typical eyes, abnormal corneal conditions (e.g., such as KC) can cause large magnitudes of HOAs that cannot be efficiently corrected by conventional optical devices (*Maeda et al., 2002*; *Pantanelli et al., 2007*). Thus, despite their habitual optical correction, KC patients are chronically exposed to severely degraded retinal inputs in their daily life. The resulting retinal image is mostly deprived of fine spatial details (i.e., high SFs) and weighted toward coarse spatial details (i.e., low SFs). It is safe to assume that this prolonged exposure to poor optical quality alters neural processing to compensate for the presence of blur. Neural compensation to image blur improves visual performance in the presence of blur but limits the benefits of improved optical correction (e.g., *Sabesan and Yoon, 2009*; *Sabesan and Yoon, 2010*; *Sawides et al., 2011*; *Sawides et al., 2010*; *Vinas et al., 2012*). When tested under full AO correction, KC patients exhibit considerably poorer VA than that predicted by optical theory or measured in observers with healthy eyes tested under similar AO-corrected optical quality (*Sabesan and Yoon, 2009*). However, little is known regarding the underlying mechanisms mediating such changes in visual processing following long-term exposure to severe optical aberrations.

Here, we used state-of-the-art AO to assess how long-term exposure (i.e., months to years) to optical defects alters neural processing of visual information in neurotypically-developed humans. To do so, we measured the impact of various amounts of optical aberrations – chronically experienced by control observers with typical, healthy eyes and KC patients – on the visual system's ability to detect contrast over a wide range of SFs under fully-corrected optical quality. KC patients differ from neurotypically-developed adults with healthy eyes based on the amounts of habitual optical aberrations experienced in their daily life, which we can bypass using AO correction to assess differences in visual processing. For well over a century, contrast sensitivity has been a valuable tool for measuring the limits of visual perception, as well as for the assessment and early diagnosis of many disorders (*Campbell and Green, 1965*; *Fechner, 1860*). The visual system is composed of SF-selective filters whose combined sensitivity determines the shape of the contrast sensitivity function (CSF) (*Blakemore and Campbell, 1969*; *Campbell and Green, 1965*; *DeValois and DeValois, 1988*). Detection of alterations in the shape of the CSF allows inferences about changes in underlying physiological processes, such as in amblyopia (*Hou et al., 2010*), autism spectrum disorder (*Guy et al., 2016*), and aging (*Yan et al., 2017*).

Given that the impact of optical aberrations is more pronounced at high SFs, we expect deficits in contrast sensitivity for high-SF signals despite correcting all optical aberrations, whereas visual sensitivity to low-SF information should remain unaffected. Neural insensitivity to fine spatial details would account for the poorer VA observed in KC patients tested under AO correction (*Sabesan and Yoon, 2009*). However, whether neural processing in KC is simply impaired for visual inputs directly affected by the eye's optics (i.e., high SFs) or undergoes a more adaptive reallocation of sensory processing resources in order to better match the degraded retinal inputs is unknown. In the present study, we provide compelling evidence that neural compensation to severe optical defects over prolonged periods of time alters visual processing across a wide range of SFs, attenuating sensitivity at high SFs but surprisingly enhancing sensitivity to low SFs. First, we show evidence of altered CSF (experiment 1) and poorer VA (experiment 2) in KC patients relative to control observers with healthy eyes, despite being tested under similar AO-corrected optical quality. Then, using an equivalent noise paradigm and a computational model of visual processing (perceptual template model [PTM]; *Bejjanki et al., 2014*; *Burgess et al., 1981*; *Dosher and Lu, 1998*; *Lu and Dosher, 1999*; *Lu and Dosher, 2008*; *Park et al., 2017*; *Pelli and Farell, 1999*), we identify the putative mechanisms underlying the changes in visual sensitivity observed in KC patients (experiment 3). The pattern of gains and losses in sensitivity in KC patients reflects SF-specific changes in signal enhancement mechanisms, with elevated and reduced internal noise levels at high and low SFs, respectively. Overall, our findings reveal that chronic exposure to poor optical quality does not result in neural deficits restricted to high SFs, but rather manifests in a functional reallocation of sensory processing resources over a wide range of SFs.

# Results

## Habitual aberrations and AO-corrected visual quality

We first need to quantify the varying amounts of habitual optical aberrations experienced by each observer before establishing that we can fully correct all optical aberrations to a similar level in both healthy and KC eyes. In total, 10 KC patients with mild-to-severe optical aberrations and 14 age-matched control observers with healthy eyes were tested monocularly (see 'Materials and methods' and *Table 1*). Habitual optical quality for the tested eye was estimated using our AO system (*Figure 2A* and 'Materials and methods') by collecting multiple wavefront measurements for each observer wearing their own, everyday optical corrective method, if any. Wavefront maps were fitted with Zernike polynomials to compute the total root mean square (total RMS; all optical aberrations) and higher-order RMS (hRMS+; HOAs only) for a 6-mm pupil (*Figure 2B* and 'Materials and methods'). Relative to the optical quality in age-matched control (CTRL) observers (mean total RMS 0.84 ± 0.26 μm, range 0.58–1.42), habitual optical quality in KC eyes (mean total RMS 3.18 ± 2.09 μm, range 1.36–7.84) remained suboptimal despite their own optical corrections (*CTRL-vs-KC total RMS*: Mann–Whitney U-test, U = 1, p<0.001, r = 0.99). Specifically, KC eyes were subject to a substantial amount of uncorrected HOAs (mean hRMS+ 1.91 ± 1.27 μm, range 0.33–4.55), much more than the healthy eyes of control observers (mean hRMS+ 0.37 ± 0.11 μm, range 0.22–0.55) (*CTRL-vs-KC hRMS+*: Welch's t-test, t = 3.83, p=0.004, d = 1.71). As previously reported (*Pantanelli et al., 2007*), KC eyes were particularly affected by large amounts of vertical coma (mean absolute $Z_7$ coefficient 1.38 ± 0.96 μm, range 0.03–2.89), much more than in typical, healthy eyes (0.12 ± 0.11 μm, range 0.02–0.34) (*CTRL-vs-KC* absolute $Z_7$ coefficient: Welch's t-test, t = 4.11, p=0.003, d = 1.84). The onset of KC in our patients was around their twenties and varied in its progression, as typically observed in this disease (*Vazirani and Basu, 2013*). As a result, the amount of habitual optical aberrations did not correlate with participant's age (Pearson correlation with habitual RMS r=+0.29, p=0.169; with habitual hRMS+ r=+0.30, p=0.155).

Consistent with previous studies from our lab (e.g., *Sabesan et al., 2007*; *Sabesan et al., 2017*; *Sabesan and Yoon, 2009*; *Sabesan et al., 2012*; *Zheleznyak et al., 2016*), our AO system allowed us to measure visual performance while effectively maintaining aberration-free optical quality, even in severe KC eyes (*Figure 2C*). To maximize AO correction during stimulus presentation, observers were trained to blink between trials and to pause if the perceptual quality got unstably poor during testing (see 'Materials and methods'). Continuous closed-loop AO correction during visual testing resulted in a residual wavefront RMS that remained not significantly different from 0.055 μm for

**Table 1.** Participant information.

| Participant | Gender | Age (years) | Total RMS (μm) | hRMS+ (μm) | AO RMS (μm) | Experiments | Habitual optical aberrations | |
|---|---|---|---|---|---|---|---|---|
| | | | | | | | *Tested eye* | *Untested eye* |
| KC1 | F | 40 | 1.49 | 0.33 | 0.050 | qCSF \| VA | Mild | Mild |
| KC2 | F | 24 | 1.36 | 0.76 | 0.046 | qCSF \| VA \| PTM | Mild | Moderate |
| KC3 | M | 27 | 1.46 | 1.14 | 0.051 | qCSF \| VA | Mild | Mild |
| KC4 | M | 43 | 1.85 | 1.34 | 0.065 | qCSF \| VA | Moderate | Moderate |
| KC5 | M | 27 | 2.12 | 1.33 | 0.082 | qCSF \| VA \| PTM | Moderate | Moderate |
| KC6 | M | 27 | 3.10 | 1.66 | 0.055 | qCSF \| VA \| PTM | Moderate | Mild |
| KC7 | F | 22 | 3.00 | 2.02 | 0.047 | qCSF \| VA | Moderate | Moderate |
| KC8 | M | 28 | 5.02 | 2.82 | 0.083 | qCSF \| VA \| PTM | Severe | Moderate |
| KC9 | M | 54 | 4.60 | 3.16 | 0.059 | qCSF \| VA \| PTM | Severe | Mild |
| KC10 | M | 55 | 7.84 | 4.55 | 0.051 | qCSF \| VA \| PTM | Severe | Mild |
| KCs (n = 10) | 3 F \| 7 M | 34.7 ± 12.4 | 3.18 ± 2.09 | 1.91 ± 1.27 | 0.059 ±0. 014 | | | |
| CTRLs (n = 14) | 3 F \| 11 M | 33.6 ± 12.3 | 0.84 ± 0.26 | 0.37 ± 0.11 | 0.051 ±0. 011 | | | |

KCs: patients with keratoconus; CTRLs: controls with healthy eyes; M: male; F: female; RMS: root mean square wavefront error computed for a 6-mm pupil size; total RMS: all habitual optical aberrations; hRMS+: higher-order aberrations; AO RMS: residual RMS under adaptive optics (AO) correction; qCSF: quick contrast sensitivity function (experiment 1); VA: visual acuity (experiment 2); PTM: perceptual template model (experiment 3).

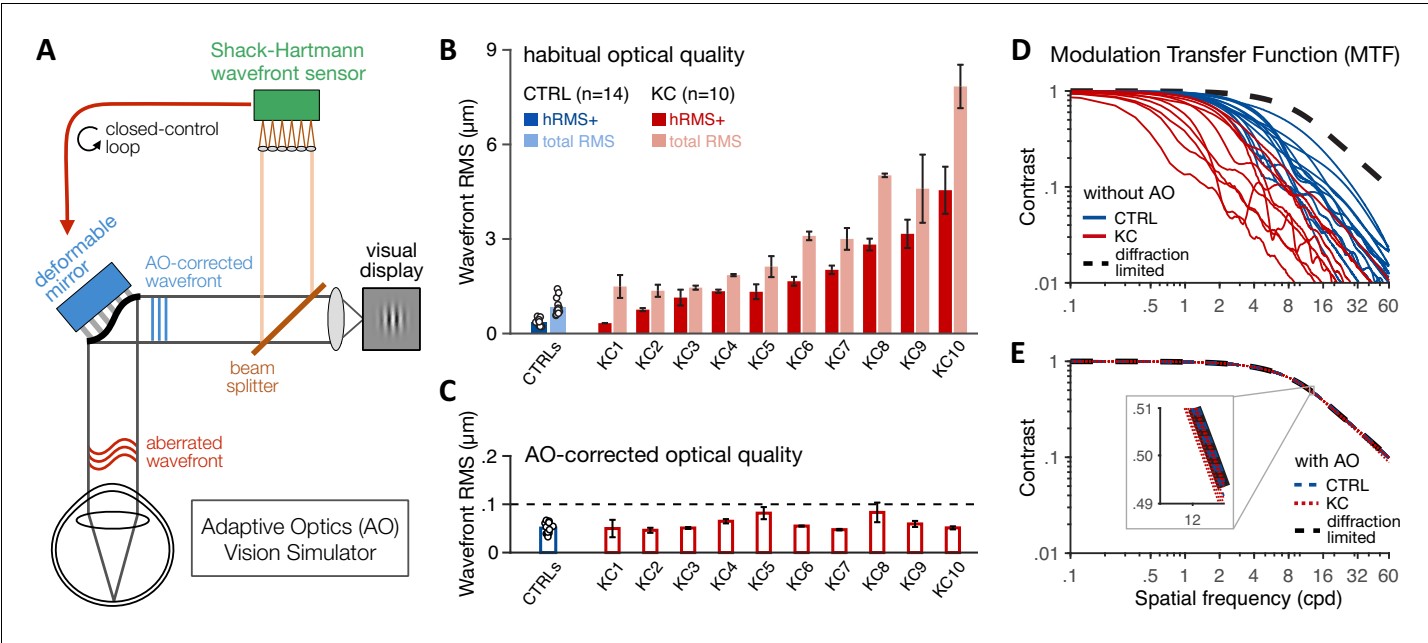

**Figure 2.** Using adaptive optics (AO) to study long-term exposure to poor optics in keratoconus (KC). (**A**) An AO vision simulator was used to maintain aberration-free image quality during visual testing by measuring the subject's aberrated wavefront using a Shack–Hartmann wavefront sensor and fully correcting it online using a deformable mirror in a closed-control loop. (**B**) Relative to age-matched control (CTRL) observers with healthy eyes, KC observers were chronically exposed to atypical amounts of habitual optical aberrations (total root mean square [RMS]), with large amounts of higher-order aberrations (hRMS+). Group-average RMS and individual data points are presented for CTRL observers (n = 14). Individual bars for each of the 10 KC patients show the habitual RMS and hRMS+ (±1 SD). (**C**) AO correction maintained similar, aberration-free optical quality during visual testing, even in severe KC eyes. Note the large difference in y-axis scale between B and C. (**D, E**) The modulation transfer function (MTF) characterizes how much contrast is transmitted by the eye's optics as a function of the spatial frequency content of the image. (**D**) Without AO correction, KC patients (red lines) were subject to severe contrast reductions relative to CTRL observers with healthy eyes (blue lines). (**E**) Under AO correction, the MTF of both KC and CTRL observers reached similar diffraction-limited optical quality, allowing us to directly measure differences in neural transfer functions between KC and CTRL observers.

The online version of this article includes the following video and figure supplement(s) for figure 2:

**Figure supplement 1.** Description of the adaptive optics vision simulator (AOVS).

**Figure supplement 2.** Examples of online adaptive optics (AO) correction in control (CTRL) and keratoconus (KC) eyes.

**Figure 2—video 1.** Online adaptive optics (AO) correction in a severe keratoconus (KC) observer (KC10).

https://elifesciences.org/articles/58734#fig2video1

both healthy eyes (mean 0.051 ± 0.011 µm, range 0.033–0.067; Student t-test: t = 1.49, p=0.161, d = 0.4) and KC eyes (mean 0.059 ± 0.014 µm, range 0.046–0.083; Wilcoxon signed-rank test: Z = 31, p=0.770, r = 0.13). More importantly, aberration-free optical quality under AO correction was similar between CTRL and KC observers (Wilcoxon rank-sum test, U = 50, p=0.259, r = 0.29), with no relation between the individual AO-corrected residual RMS and the amount of habitual optical aberrations without AO correction (Pearson correlation with habitual RMS r=+0.30, p=0.155; with habitual hRMS+ r=+0.33, p=0.120).

The modulation transfer function (MTF) of the eye's optics characterizes the reduction in signal contrast of individual SF contents imaged on the retina (*Figure 2D, E*). The reduction in contrast is pronounced at mid-to-high SFs, particularly in KC eyes due to the atypical levels of uncorrected blur (*Figure 2D*). Importantly, AO correction allowed us to correct all optical aberrations and maintain similar, diffraction-limited optical quality in all participants (*Figure 2E*). Thus, online AO correction effectively and continuously provided nearly perfect optical quality during visual testing, allowing us to bypass optical factors and assess differences in neural transfer function between CTRL with healthy eyes and KC patients chronically exposed to atypical amounts of optical aberrations.

# Experiment 1: altered CSF following long-term exposure to optical defects

First, we assessed the CSF in both KC patients (N=10) and age-matched control observers (N=14) under aberration-free conditions. Observers performed an orientation discrimination task at fixation (*Figure 3A*) in which they reported the orientation of ±45°-tilted Gabor patches varying in both contrast and SF (in cycles per degree [cpd]). To optimize data collection, we used the *quick CSF* method (*Hou et al., 2010*; *Hou et al., 2016*; *Lesmes et al., 2010*), which combines Bayesian adaptive inference with a trial-by-trial information gain strategy to estimate the observer's CSF as a truncated log-parabola with four parameters (*Figure 3B* and 'Materials and methods'): (1) peak sensitivity, $CS_{max}$; (2) peak frequency, $SF_{peak}$; (3) bandwidth, $\beta$; and (4) low-SF truncation level, $\delta$. The low-SF truncation level determines sensitivity at low SFs ($CS_{low}$). We also estimated the high-SF cutoff ($SF_{cutoff}$) from qCSF fits – a measure of VA. The qCSF method has been used to characterize CSF in both neurotypical and diverse clinical populations (*Hou et al., 2010*; *Hou et al., 2016*; *Lesmes et al., 2010*; *Thurman et al., 2016*; *Yan et al., 2017*).

Under similar, aberration-free optical quality, KC observers exhibited altered CSF relative to control observers (*Figure 4A*), showing impairments for high SFs (for SFs $\geq$ 8.13 cpd) but also improvements for low SFs (for SFs $\leq$ 0.92 cpd). In other words, even when both groups had matched retinal image quality, there were still substantial differences in contrast sensitivity. This pattern of gains and losses changed the shape of the CSF, revealing both impaired sensitivity at high SFs ($SF_{cutoff}$; CTRL: 37.5 cpd [35.1–40.3; 95% CI]; KC 29.0 cpd [27.7–31.2]; p<0.001) and better sensitivity at low SF ($CS_{low}$; CTRL 4.60 [3.74–5.46]; KC 7.41 [5.82–8.62]; p<0.001). Improved sensitivity at low SFs reflected a reduction in the low-SF truncation level in KC ($\delta$; CTRL +0.063 [+0.034 +0.101]; KC −0.021 [−0.055 +0.028]; p=0.0025). Moreover, the impairment in high-SF cutoff was marginally correlated with the reduction in low-SF truncation across observers (r=+0.38, p=0.0675). The overall result was a shift in peak SF toward lower SFs in KC ($SF_{peak}$; CTRL 3.36 cpd [3.25–3.72]; KC 2.70 cpd [2.54–2.91]; p<0.001). Notably, these CSF changes were roughly balanced, resulting in the area under the CSF (AUCSF) being similar between groups (CTRL 2.63 [2.60–2.73]; KC 2.66 [2.60–2.73]; p=0.461). This is also consistent with the fact that we observed no significant changes in amplitude ($CS_{max}$;

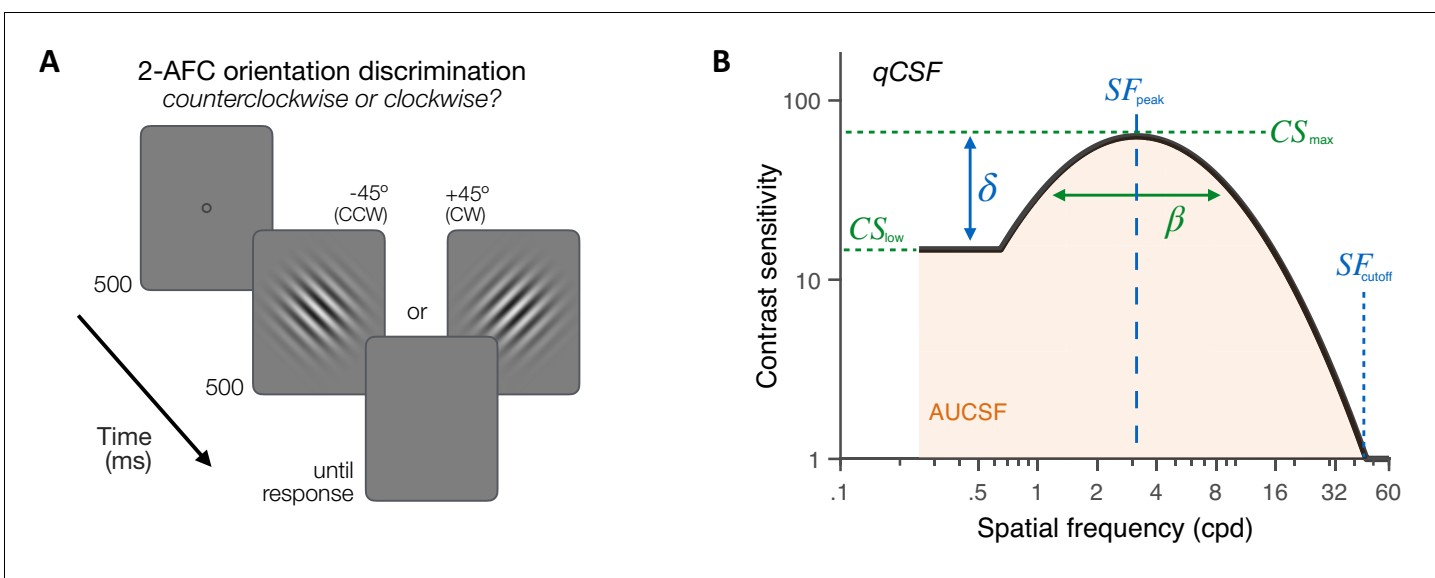

**Figure 3.** Experiment 1: contrast sensitivity function (CSF) measurement. (**A**) *Stimuli, task and timeline*. Each trial began with a dynamic fixation point. After a blank screen, a ±45°-oriented Gabor stimulus was presented at fixation using a 500-ms temporal Gaussian envelope. Stimuli varied in contrast and spatial frequency (SF) from trial to trial. Observers judged whether the stimulus was tilted ±45° from vertical using a 2 alternative forced choice (AFC) task. Full adaptive optics correction was maintained during testing. (**B**) *Quick CSF method*. The qCSF estimates the CSF through four parameters: peak sensitivity ($CS_{max}$), peak SF ($SF_{peak}$), bandwidth ($\beta$), and low-SF truncation ($\delta$). The area under the CSF (AUCSF), low-SF sensitivity ($CS_{low}$), and high-SF cutoff ($SF_{cutoff}$) were also estimated from qCSF fits.

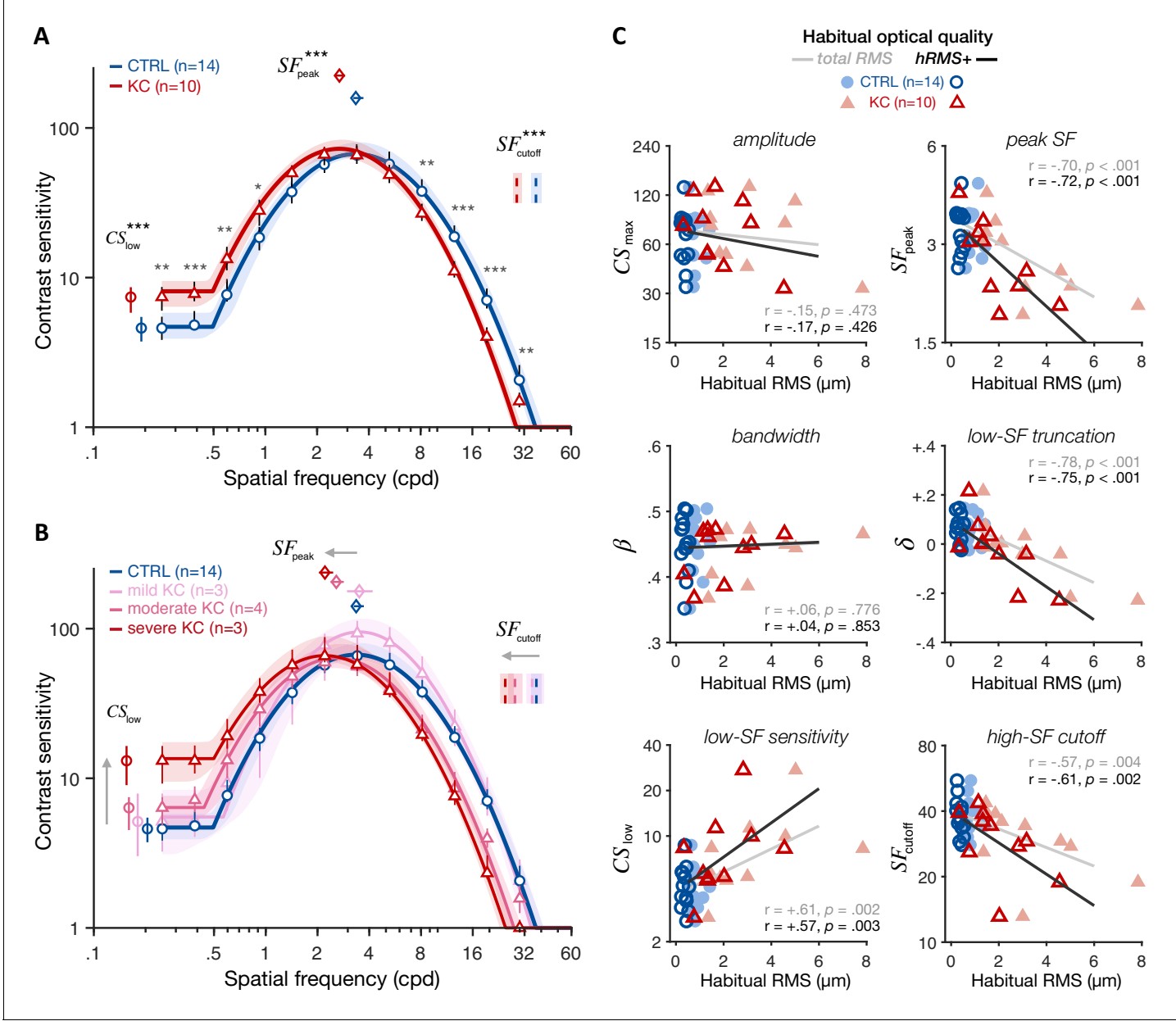

**Figure 4.** Experiment 1: altered contrast sensitivity function (CSF) following long-term exposure to poor optics. All results are for visual images fully corrected using adaptive optics (AO), for both keratoconus (KC) and age-matched control (CTRL) observers. (**A**) qCSF results. Relative to control observers (blue curve), KC patients (red curve) showed altered CSFs that were shifted toward lower SFs, with both impaired high-SF sensitivity and improved low-SF sensitivity. Shaded areas and error bars represent bootstrapped 95% confidence intervals (CIs). Asterisks indicate significant differences computed from bootstrapping between groups (*p<0.05; **p<0.01; ***p<0.001). Comparisons at each of the 12 SFs were corrected for multiple comparisons using Bonferroni correction. (**B**) Same as **A** but with KC patients divided into separate groups based on KC severity for illustration purposes (see also **Figure 4—figure supplement 1**). (**C**) Link between CSF and habitual optical quality. Altered CSF under AO correction correlated with the amount of habitual optical aberrations of each subject. Each panel shows individual parameter estimates plotted as a function of each subject's habitual root mean square (RMS; total RMS: light filled symbols; hRMS+: darker open symbols). Linear regression fits and Pearson correlation coefficients are plotted for each RMS measure.

The online version of this article includes the following figure supplement(s) for figure 4:

**Figure supplement 1.** Experiment 1: altered contrast sensitivity function (CSF) as a function of KC severity.

**Figure supplement 2.** Experiment 1: altered contrast sensitivity function (CSF) does not depend on participant's age.

CTRL 67.00 [63.1–80.3]; KC 72.5 [63.8–83.5]; p=0.349) or bandwidth ($\beta$; CTRL 2.82 [2.62–2.92]; KC 2.75 [2.60–2.89]; p=0.390).

Critically, both gains and losses in contrast sensitivity under aberration-free conditions were more pronounced in KC patients who experienced large amounts of optical aberrations in their everyday life (*Figure 4B* and *Figure 4—figure supplement 1*). Note that KC patients were split based on their habitual optical aberrations only for illustration purposes. Changes in qCSF parameters under full AO correction correlated with the amount of habitual aberrations each observer experienced in their everyday life (*Figure 4C*; Pearson correlation coefficients using hRMS+; $SF_{peak}$: r=−0.72, p<0.001; $SF_{cutoff}$: r=−0.61, p=0.002; $\delta$: r=−0.75, p<0.001; $CS_{low}$: r=+0.57, p=0.003). No correlation was observed for other qCSF parameters (i.e., $CS_{max}$, $\beta$, or AUCSF). A similar pattern was observed using the total RMS as an index of habitual optical quality (*Figure 4C*). Moreover, the age at disease onset in our patients was around their twenties, as typically observed in KC (*Vazirani and Basu, 2013*). Participant's age is therefore a good proxy for the exposure duration to the optical aberrations caused by KC. Yet, none of the changes in CSF observed in KC patients actually correlated with participant's age (*Figure 4—figure supplement 2*). There was a tendency for the AUCSF and amplitude ($CS_{max}$) to decrease with participant's age, consistent with the overall reduction in sensitivity with age (e.g., *Yan et al., 2017*). Note that substantial changes in visual processing with age are typically observed in much older (>60 years old) observers (e.g., *Derefeldt et al., 1979*; *Elliott et al., 1995*; *Yan et al., 2017*).

## Experiment 2: poorer VA following long-term exposure to optical defects

VA measurements using high-contrast Snellen E letters (*Figure 5A* and 'Materials and methods') showed that both groups had better than 20/20 vision when tested under full AO correction, as expected. However, KC patients (15.6 ± 1.9 Snellen VA, range 13.4–18.5) had poorer VA than control observers (12.6 ± 1.3 Snellen VA, range 10.6–15.5) (*Figure 5B*; *CTRL-vs-KC logMAR VA*: t = 4.68, p<0.001, d = 1.94). This deficit in high-contrast VA under full AO correction is consistent with previous findings (*Sabesan and Yoon, 2009*) and correlated with the participant's amount of habitual aberrations experienced outside the AO (*Figure 5C*; total RMS: r=+0.74, p<0.001; hRMS+: r=+0.79, p<0.001), but not with participant's age (*Figure 5—figure supplement 1*; r=+0.24, p=0.250). Moreover, VA deficits under AO correction correlated with the changes in qCSF parameters (*Figure 5D*), such as peak SF (r = −0.73, p<0.001), high-SF cutoff (r = −0.74, p<0.001), and notably low-SF truncation (r = −0.66, p<0.001) and low-SF sensitivity (r=+0.42, p=0.039). That is, poorer VA in KC under AO correction was predicted by both impaired high-SF sensitivity as well as improved low-SF sensitivity. Altogether, these findings indicate that long-term exposure to severe optical defects alters the CSF in a way that results in impaired fine spatial vision, but also in better sensitivity to coarser spatial information that is less affected by the eye's optics. This pattern of results is suggestive of a functional reallocation of sensory processing resources across a broad range of SFs that is expected to optimize visual processing of severely aberrated retinal images, but limits perceptual quality under fully-corrected optical quality.

## Experiment 3:altered internal additive noise levels following long-term exposure to optical defects

To better understand how long-term exposure to optical defects alters visual processing across SFs, we used an equivalent noise paradigm (*Bejjanki et al., 2014*; *Burgess et al., 1981*; *Dosher and Lu, 1998*; *Lu and Dosher, 1999*; *Lu and Dosher, 2008*; *Park et al., 2017*) in which we measured perceptual thresholds as a function of varying external noise levels (see 'Materials and methods'). Six KC patients and eight age-matched control observers (see 'Materials and methods' and *Table 1*) performed an orientation discrimination task under varying levels of dynamic white noise added to the stimuli (*Figure 6A*), judging whether a foveally presented grating was tilted ±45° from vertical. Five different SFs (0.5, 1, 3, 9, and 16 cpd) were tested across different experimental sessions. Stimulus presentation was controlled using the *FAST* method (*Vul et al., 2010*), an advanced adaptive psychophysical technique that allowed us to estimate relevant model parameters in just 480 trials per stimulus SF and per observer. Changes in perceptual thresholds with external noise result in a characteristic threshold-versus-noise (TvN) function, which can be fitted with the PTM (*Dosher and*

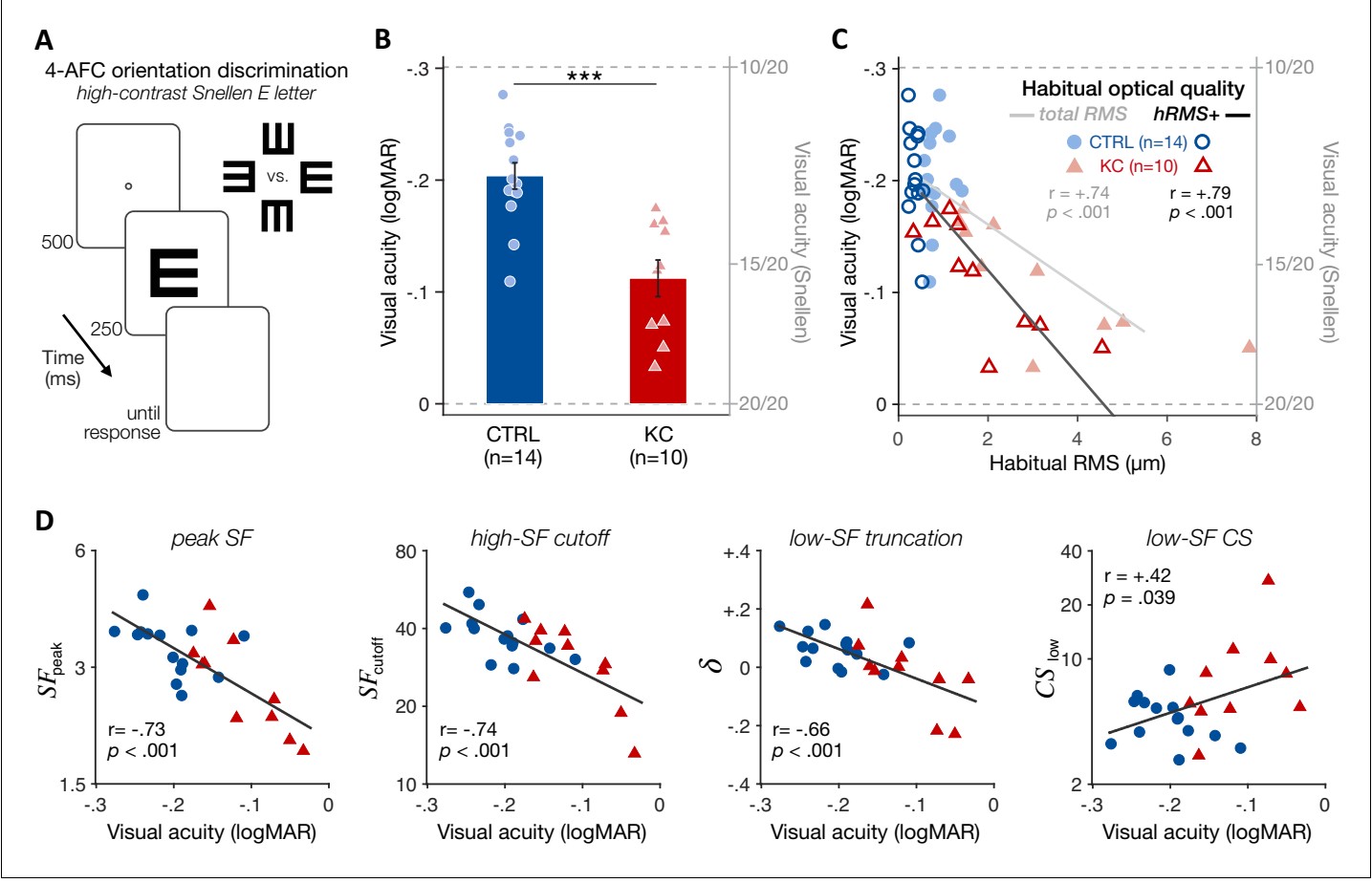

**Figure 5.** Experiment 2: poorer high-contrast visual acuity (VA) following long-term exposure to poor optics. (**A**) VA acuity thresholds were measured under adaptive optics (AO) correction using a four alternative forced choice (AFC) discrimination task. The size of high-contrast, Snellen E letter stimuli varied from trial to trial to estimate 62.5%-correct VA thresholds (in logMAR). (**B**) Relative to age-matched control (CTRL) observers, keratoconus (KC) patients showed poorer VA despite being tested under similar aberration-free optical conditions. Error bars correspond to ±1 SEM, with each data point representing an individual observer (CTRL: blue circles; KC: red triangles). (**C**) Poorer habitual optical quality was associated with stronger VA deficits under aberration-free conditions; total root mean square (RMS): all habitual aberrations; hRMS+: higher-order aberrations. (**D**) Deficits in VA under AO correction correlated with the changes in qCSF parameters observed in experiment 1. Linear regression (solid line) and Pearson correlation coefficients are indicated in **C** and **D**.

The online version of this article includes the following figure supplement(s) for figure 5:

**Figure supplement 1.** Experiment 2: poorer VA does not depend on participant's age.

*Lu, 1998*; *Lu and Dosher, 1999*; *Lu and Dosher, 2008*) to quantify the effects of noise on perception (*Figure 6*). The PTM is a computational model of visual processing that had been successfully used to identify the mechanisms underlying perceptual differences for a wide range of brain functions, such as attention (*Dosher and Lu, 2000*; *Lu and Dosher, 1998*) and perceptual learning (*Dosher and Lu, 1998*), as well as between specific populations, such as in amblyopia (*Levi and Klein, 2003*), autism (*Park et al., 2017*), dyslexia (*Sperling et al., 2005*), and cortical blindness (*Cavanaugh et al., 2015*).

The PTM considers that the differences in perceptual performance between groups can result from changes in three separate mechanisms (*Figure 6*): (1) signal enhancement, (2) signal selectivity, or (3) gain control (nonlinearity change). Each signal processing mechanism is implemented within the PTM observer model by changes in one of three possible sources of noise: (1) internal additive noise ($A_{add}$), (2) external noise filtering ($A_{ext}$), and (3) multiplicative internal noise ($A_{mul}$). Signal enhancement improves the ratio of the signal relative to the internal additive noise. Elevated internal additive noise indicates impaired signal enhancement mechanisms and leads to worse perceptual

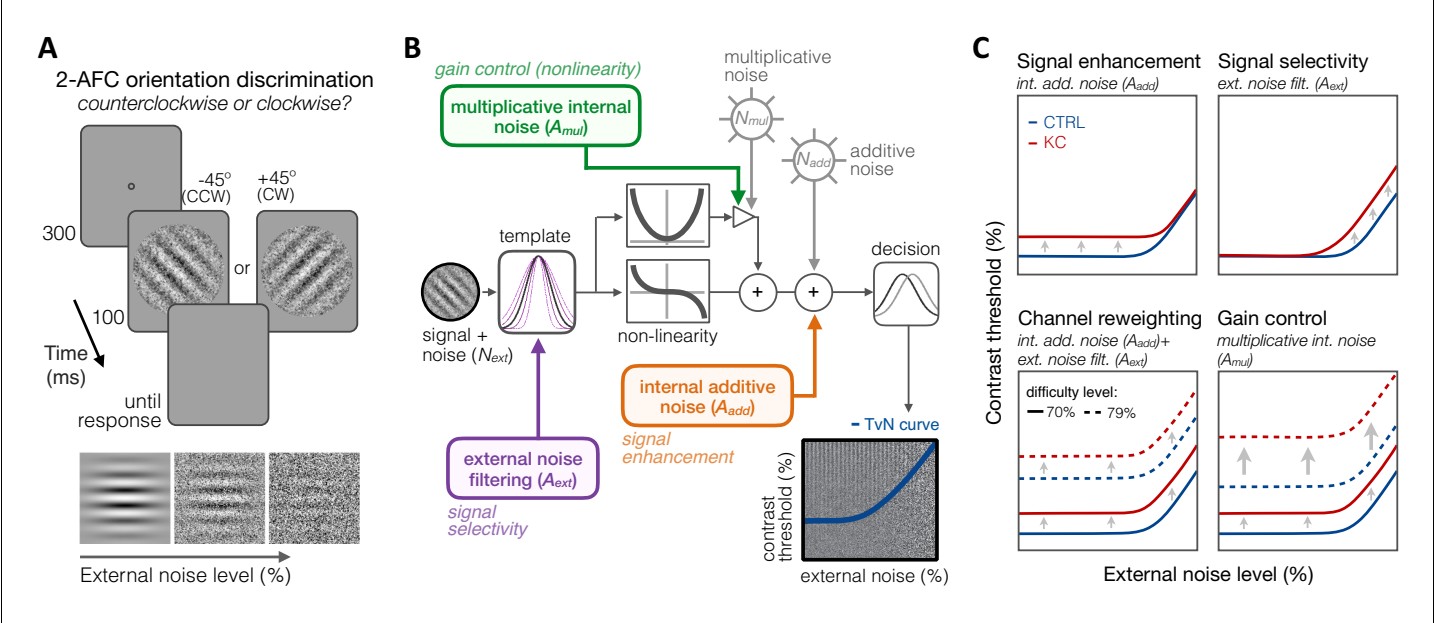

**Figure 6.** Experiment 3: equivalent noise paradigm and perceptual template model (PTM). (**A**) *Stimuli, task, and timeline.* In each trial, an oriented grating embedded in dynamic Gaussian pixel noise was presented. Observers judged whether stimuli were tilted ±45° from vertical. (**B**) *Schematic representation of the PTM.* This model consists of five main components: a perceptual template processing signals embedded in external noise ($N_{ext}$), a nonlinear transducer function, two internal noise sources (multiplicative $N_{mul}$ and additive $N_{add}$), and a decision process. The output of the decision process models limitations in sensitivity as equivalent internal noise, with threshold-vs-noise (TvN) curves having a characteristic nonlinear shape. At low external noise, internal noise dominates and added external noise has little effect, resulting in the TvN curve's flat segment. Once external noise level exceeds that of internal noise, stronger signal contrast is needed to overcome added external noise, resulting in the TvN curve's rising segment. Adapted from *Lu and Dosher, 2008*. (**C**) *Predictions.* Distinct mechanisms can account for group differences between control (CTRL) and keratoconus (KC) participants. (1, *upper left*) Elevated internal additive noise ($A_{add}$) yields increased thresholds at the TvN curve's flat portion, reflecting differences in signal enhancement mechanisms.(2, *upper right*) Poorer external noise filtering ($A_{ext}$) limits thresholds at the TvN curve's rising portion, reflecting differences in signal selectivity (tuning). (3, *lower left*) A combination of the two results in increased thresholds across external noise levels and difficulty levels, and suggests channel reweighting. (4, *lower right*) Elevated multiplicative internal noise ($A_{mul}$) results in a similar pattern across external levels but the differences scale with difficulty levels (i.e., signal contrast), acting similarly to a gain control mechanism (*nonlinearity change*).

thresholds at low external noise levels (*Figure 6C*, *upper left*). No difference is observed when high external noise is the primary limiting factor as both the signal and the external noise in the stimulus are amplified. Signal selectivity (tuning) determines the capacity of the perceptual template to filter out external noise, which affects performance only at high external noise levels (*Figure 6C*, *upper right*). A combined effect of higher internal additive noise and poorer external noise filtering elevates thresholds at all external noise levels, similarly across difficulty levels, which suggests channel reweighting (*Figure 6C*, *lower left*). Finally, higher multiplicative internal noise ($A_{mul}$) would also impair thresholds at all external noise levels, but via a non-uniform shift in TvN functions across difficulty levels (*Figure 6C*, *lower right*). The magnitude of multiplicative internal noise is proportional to stimulus contrast: Higher $A_{mul}$ results in greater elevation in internal noise with higher signal contrast, acting similarly to a gain control mechanism (i.e., compressive response at higher contrasts).

First, we analyzed the data using the conventional PTM (*Dosher and Lu, 1998*; *Lu and Dosher, 1999*; *Lu and Dosher, 2008*) to characterize the relative difference in noise-limiting factors between the CTRL and KC groups (see 'Materials and methods'). To do so, contrast thresholds were estimated for each observer as a function of external noise, difficulty level, and stimulus SF. Then, we fitted the data with the PTM in a conventional manner using a least-squares procedure (*Figure 7*). Given the large individual variability in KC habitual aberrations and its impact on the CSF, KC participants were split into two groups based on KC severity using a median split (total RMS 3.85 μm; hRMS+ 2.24 μm), resulting in three groups based on habitual optical quality: age-matched controls with healthy eyes (N = 8; total RMS 0.88 ± 0.28 μm; hRMS+ 0.34 ± 0.12 μm), mild/moderate KC (N = 3; total RMS 2.19 ± 0.87 μm; hRMS+ 1.25 ± 0.46 μm), and severe KC (N = 3; total RMS

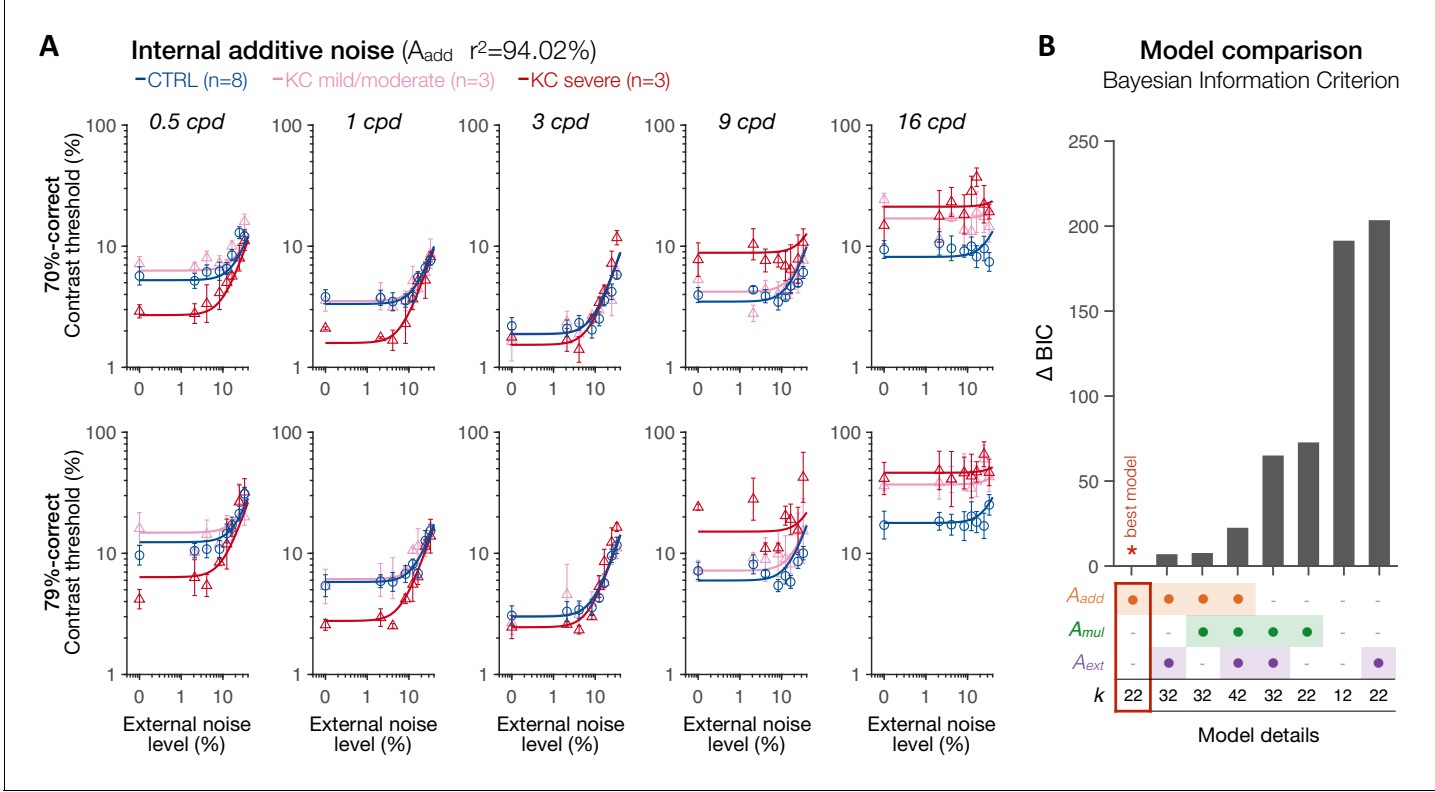

**Figure 7.** Experiment 3: spatial frequency (SF) specific changes in internal additive noise mediate contrast sensitivity differences following long-term exposure to optical defects. (**A**) Contrast thresholds were measured under full adaptive optics (AO) correction as a function of external noise levels for five different SFs (0.5, 1, 3, 9, and 16 cpd) and for 70.71%-correct (*upper row*) and 79.37%-correct (*lower row*) difficulty levels. Relative to control (CTRL) observers, severe keratoconus (KC) patients showed impaired thresholds at high SFs and better thresholds at low SFs. These group differences were observed at low, but not at high, external noise levels, and were less pronounced in KC patients with mild-to-moderate amounts of habitual optical aberrations. Data were fitted with the perceptual template model to evaluate the contribution of distinct sources of inefficiencies: internal additive noise ($A_{add}$), multiplicative internal noise ($A_{mul}$), external noise filtering ($A_{ext}$), or any combination of these factors. The best model explaining the differences between groups was the internal additive noise model. (**B**) *Model comparisons.* Bayesian information criterion (BIC) evidence was computed for each model to identify which model best explained the data while penalizing for a greater number of free parameters (*k*). The value of the best model was subtracted from all BIC values (ΔBIC). The best model (ΔBIC = 0) was the model assuming solely changes in internal additive noise across SFs between CTRL and KC groups.

The online version of this article includes the following figure supplement(s) for figure 7:

**Figure supplement 1.** Experiment 3: results of the full model and of the internal additive noise model.

**Figure supplement 2.** Experiment 3: results of the two mixture models with internal additive noise.

**Figure supplement 3.** Experiment 3: results of the multiplicative internal noise + external noise filtering model and multiplicative internal noise model.

**Figure supplement 4.** Experiment 3: results of the external noise filtering model and of the null model.

**Figure supplement 5.** Experiment 3: independent analysis for low spatial frequencies (SFs) supports the internal additive noise model.

**Figure supplement 6.** Experiment 3: independent analysis for high spatial frequencies (SFs) supports the internal additive noise model.

**Figure supplement 7.** Experiment 3: contrast sensitivity function at low and high external noise levels.

5.82 ± 1.76 μm; hRMS+ 3.51 ± 0.91 μm). All the data from all five SFs were included in the analysis, with the model estimating separate TvNs for each group, each SF, and each difficulty levels (see 'Materials and methods'). The resulting TvN curves estimated by the PTM exhibited characteristic nonlinear patterns (*Figure 7*).

To assess the mechanisms underlying the differences in contrast sensitivity between groups, several variants of the PTM were fit to the data: no group difference (null model), changes in internal additive noise ($A_{add}$), external noise filtering ($A_{ext}$), or multiplicative internal noise ($A_{mul}$), mixtures of them ($A_{add} + A_{mul}$; $A_{add} + A_{ext}$; $A_{mul} + A_{ext}$), and a full model assuming changes in all three mechanisms ($A_{add} + A_{ext} + A_{mul}$). To account for the large differences in the number of free parameters across models, we computed the Bayesian information criterion (BIC) evidence that determines

which model best fits the data while penalizing for a greater number of free parameters. *Figure 7A* shows the results of the best model (see *Figure 7—figure supplements 1–4* for the results from each of the eight possible variants of the PTM). Overall, the differences in contrast thresholds between groups were best explained by a simple model assuming solely SF-dependent changes in internal additive noise (*Figure 7B*; $A_{add}$; $r^2 = 94.02$), followed by mixture models that included internal additive noise as one of the free parameters, and then by the full model. Other models that did not allow changes in internal additive noise ($A_{add}$) could not account for the differences between groups. Relative to the control observers, severe KC patients showed a reduction in internal additive noise ($A_{add}$) by −49.2% and −53.1% at 0.5 and 1 cpd, respectively. Mild/moderate KC patients did not show such benefits at low SFs, but rather a slight elevation in internal noise by +20.4% and +5.7% at 0.5 and 1 cpd, respectively. At 3 cpd, mild/moderate KC observers showed a negligible difference in internal additive noise (+1.1%) relative to control observers, whereas severe KC patients still showed a slight reduction in internal additive noise (−18.7%). At high SFs, all KC observers exhibited poorer thresholds due to large elevations in internal additive noise, by +21.2% and +110.8% at 9 and 16 cpd for mild/moderate KCs, and by +158.4% and +165% at 9 and 16 cpd for severe KC patients.

Additional PTM analyses, separately for low and high SFs (*Figure 7—figure supplements 5* and *6*), further supported SF-specific changes in internal additive noise as the primary mechanism underlying altered CSF following long-term exposure to poor optical quality. Furthermore, these results replicate the pattern of gains and losses in contrast sensitivity found in experiment 1 (*Figure 7—figure supplement 7*), despite differences between the two experiments (e.g., high uncertainty regarding the SF of the upcoming target stimulus on a given trial in experiment 1, whereas there was no uncertainty in experiment 3). When almost no external noise was added to the target stimuli, observers with severe KC showed higher contrast sensitivity for low SFs and impaired sensitivity for high SFs. This pattern was less pronounced in mild/moderate KC participants. At high external noise levels, contrast sensitivity was more comparable across groups, consistent with the predictions of the internal additive noise model (*Figure 6C*). Note that the differences at high SFs under high external noise (*Figure 7—figure supplement 7*) are likely due to the fact that the nonlinearity characteristic of TvN curves was not well captured and internal additive noise remained the main limiting factor even at the highest external noise level. Altogether, these results strongly support changes in internal additive noise (i.g., signal enhancement) as the primary mechanism underlying the differences in contrast sensitivity at both low SFs and high SFs in KC patients.

To better assess individual differences in the underlying sources of altered visual processing across SFs and relate them to each observer's habitual optical quality, we estimated TvN curves and parameter estimates for each individual participant using a hierarchical Bayesian model (see 'Materials and methods'). By assuming that the variability between participants follows a population-level distribution, this method allowed us to estimate group-level 'traits' in terms of signal processing given the presence of external noise. We then tested whether the level of internal additive noise, as well as other individual PTM estimates, correlated with the amount of habitual optical aberrations experienced by each participant (*Figure 8*). Consistent with the SF-specific changes in sensitivity observed in KC patients, poorer habitual RMS was associated with reduced individual internal additive noise levels at low SFs (0.5 and 1 cpd) (*Figure 8A*; total RMS: r = −0.65, p=0.011; hRMS+: r = −0.63, p=0.016). At high SFs, individual noise estimates were more variable, particularly at 16 cpd where we could not reliably estimate the nonlinear segment of individual TvN curves. Thus, we restricted this analysis to individual noise estimates from the 9 cpd condition. We found that poorer habitual optical quality was positively correlated with elevated individual internal additive noise levels at high SFs (*Figure 8A*; total RMS: r=+0.65, p=0.012; hRMS+: r=+0.70, p=0.005). No correlation was found at low or high SFs with either multiplicative internal noise or external noise filtering (*Figure 8B, C*), supporting the finding that SF-specific changes in internal additive noise alone can account for the impact of long-term exposure to poor optical quality on visual processing.

Taken together, the present results reveal that long-term exposure to severe optical defects causes a broad functional reallocation of sensory processing resources across a wide range of SFs, which is mediated by SF-specific changes in signal enhancement mechanisms of SF-selective neurons.

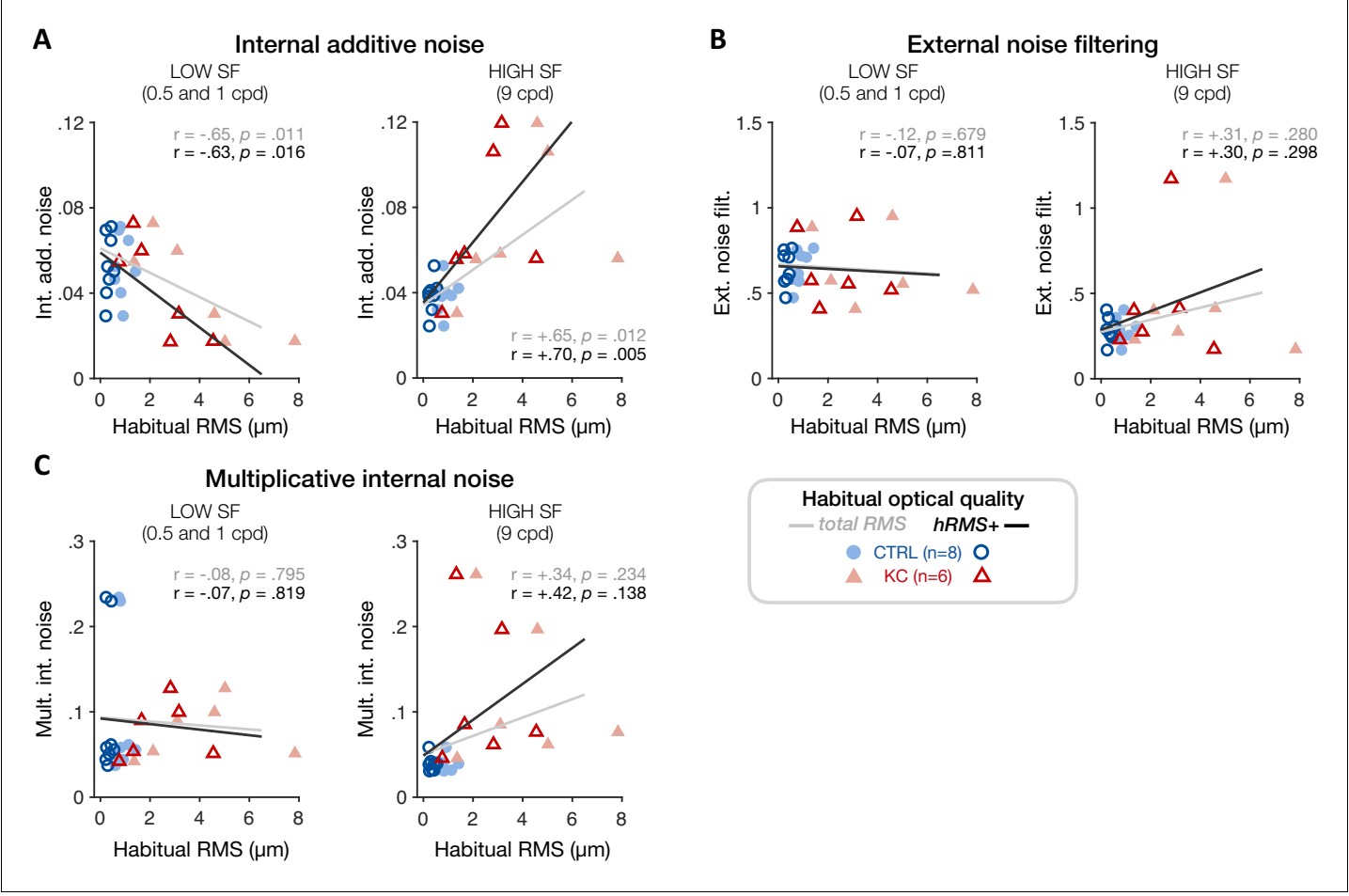

**Figure 8.** Experiment 3: correlation between individual noise estimates and participants' habitual optical quality at low and high spatial frequencies (SFs). (**A**) Poorer habitual optical quality was associated with SF-specific changes in internal additive noise, with reduced internal additive noise at low SFs (*left panel*) and elevated internal additive noise at high SFs (*right panel*). Other noise estimates (**B**: external noise filtering; **C**: multiplicative internal noise) did not correlate with habitual optical quality, consistent with internal additive noise being the primary source of inefficiency mediating the effects of long-term exposure to poor optical quality. Data points correspond to individual estimates (CTRL: control, blue circles; KC: keratoconus, red triangles) plotted as a function of the participant's habitual root mean square (RMS) wavefront error (i.e., total RMS: all habitual aberrations; hRMS +: higher-order aberrations).

## Discussion

The present study combined advanced optics, psychophysical, and computational methods to assess neural compensation mechanisms in response to long-term (i.e., from several months up to several decades) exposure to poor optical quality. An AO vision simulator was used to fully bypass optical limitations and directly access and isolate visual processing in human adults who are otherwise exposed to various amounts of optical aberrations. Under AO correction, both control and KC observers can be considered as a single group of neurotypically-developed adults chronically exposed to varying levels of uncorrected optical aberrations in their daily life. Our results provide evidence of a broad, functional reallocation of sensory processing resources across a wide range of SFs following long-term exposure to severely degraded retinal inputs. Under fully-corrected optical conditions, KC patients showed a loss in sensitivity to high-SF information, which was due to elevated internal additive noise (i.e., impaired signal enhancement) of high-SF selective filters. Moreover, severe KC patients showed enhanced sensitivity to low-SF information due to reduced internal additive noise (i.e., improved signal enhancement) of low-SF selective filters. Notably, the magnitudes of low-SF enhancements and high-SF impairments were correlated across observers, and both effects were more pronounced in severe KC cases. This pattern of gains and losses in visual

sensitivity reveals fundamental properties of adaptive neural mechanisms compensating for the chronic exposure to blurred retinal images.

A key function of sensory systems is to adapt to our sensory environment. Contrast sensitivity represents the foundation for the brain's coding of visual information and is central for characterizing both visual functions and clinical disorders. The shape of the CSF likely reflects the visual system's sensitivity to stimulus properties that are useful for perception. In typical observers with healthy eyes, the drop in sensitivity at high SFs is mostly due to blurring from the eye's optics (*Banks et al., 1987*; *Campbell and Green, 1965*). Optical blur reduces the strength and reliability of high-SF retinal inputs, leading to weak and unreliable neural responses to fine spatial details. The drop in signal-to-noise ratio at high-SFs has also been linked to a stronger response to noise with increasing frequency rather than to a change in spatial sensitivity per se (*Brady and Field, 1995*). In KC, optical aberrations become approximately six times worse compared to typical levels (*Pantanelli et al., 2007*), strongly attenuating high-SF retinal signals and depriving the visual system from fine spatial information. Our findings show that the severe and prolonged exposure to optically degraded retinal inputs under natural viewing results in neural insensitivity to fine spatial details, even when viewing images that are near-perfect optically. Using the PTM, we found that this neural insensitivity to high-SF signals was due to elevated internal additive noise levels (i.e., impaired signal enhancement) of high-SF filters. Internal noise impairs the reliability of sensory representations and is a key factor in perceptual variability within the nervous system (*Osborne et al., 2005*). Increased internal noise levels of high-SF filters could account for the fact that when tested under AO correction, KC patients showed considerably poorer letter acuity than control observers with typical optical quality whose AO-corrected acuity approaches the limit imposed by photoreceptor sampling.

More importantly, chronic exposure to degraded high-SF inputs did not solely induce impairments in high-SF contrast sensitivity and high-contrast letter acuity under AO correction, but was also associated with better-than-normal sensitivity to low-SF signals. This improvement in contrast sensitivity at low SFs was mediated by a reduction in internal additive noise levels (i.e., improved signal enhancement) of low-SF selective filters, and correlated with the deficits in VA. Thus, the effects of long-term exposure to poor ocular optics cannot be simply characterized as deficits at high SFs, but rather as altered visual sensitivity across a broad SF range via changes in signal enhancement mechanisms that attenuate high-SF signals while enhancing low-SF information. This pattern of gains and losses in sensitivity can be understood by considering how habitual optics of KC eyes affect retinal image quality. While optical blur strongly degrades high-SF signals, lower SFs are less compromised by blur and become predominant and more reliable than higher SFs, especially under large amounts of blur. As a result, KC patients are in a low-pass state of adaptation during natural vision, being deprived of high-SF information and having to rely more on low-SF information to interact with their environment.

Adaptation acts to maintain sensory systems operating within the limited dynamic range afforded by the brain's limited resources (*Ghodrati et al., 2019*). In this context, the CSF reflects an optimal allocation of the brain's limited sensory resources to process a wide range of spatial details. The pattern of gains and losses in sensitivity we observed could reflect neural compensation mechanisms that optimize visual processing to the structure of the degraded retinal inputs. For instance, short-term adaptation to different stimulus speeds alters the spatiotemporal CSF over a broad range of spatial and temporal frequencies, optimizing visual sensitivity to the new environment (*Gepshtein et al., 2013*). Physiologically, cells in the primary visual cortex (V1) adaptively change their responses to match the input properties of natural (broadband) stimulation and enhance information transmission in visual processing (*Sharpee et al., 2006*). Moreover, brief adaptation to a single stimulus can reorganize neuronal correlations across an entire network, resulting in specific, stimulus-dependent changes in the population code's efficiency (*Gutnisky and Dragoi, 2008*). What is unknown, however, is what kinds of neural changes occur in adult visual systems during years of adaptation periods, as experienced by participants with severe optical defects such as KC patients. We conjecture that long-term exposure to severely degraded retinal image quality results in a reallocation of the brain's limited sensory resources, improving visual processing in KC patients across many stimuli and tasks. Indeed, whereas KC participants suffer from poorer acuity under full AO correction, they actually show substantially better acuity under their own optical quality than typical observers tested under the same degraded optical conditions (*Sabesan and Yoon, 2010*). It is worth noting that the changes in contrast sensitivity we observed do not necessarily imply similar

changes in suprathreshold perception. Unlike contrast sensitivity, the perceived contrast of gratings above threshold is largely independent of SF (*Georgeson and Sullivan, 1975*). Contrast constancy is mediated by contrast gain adjustment mechanisms that compensate for earlier attenuation in sensitivity and achieve a 'deblurring' of the retinal image (*Georgeson and Sullivan, 1975*). Although we did not directly assess suprathreshold contrast perception, there is evidence that contrast constancy is tightly linked to the eye's optics. For instance, astigmatic observers show suprathreshold compensation for the orientation-specific neural deficit in contrast sensitivity caused by their uncorrected blur (*Georgeson and Sullivan, 1975*). Thus, chronic exposure to severe amounts of optical aberrations might also impact suprathreshold perception in severe KC participants.

All KC patients included in our study reported an onset of the KC-induced optical aberrations around their twenties, which is typical in this disease (*Vazirani and Basu, 2013*). KC progression can vary from one individual to another, usually stabilizing after a few years. We found that individual differences in visual processing under AO correction were well accounted for by the amount of habitual optical aberrations experienced by each observer, regardless of participant's age. Thus, neural compensation to the eye's optics was primarily driven by the severity of uncorrected optical aberrations rather than by the number of years exposed to uncorrected blur per se. Of note, all the participants included in our study had stable optical quality for at least 1 year before testing. This allows us to conclude that 1 year of sustained exposure to poor optics is sufficient to cause the results reported here. Whether shorter periods of exposure to poor optics – months or even weeks – are sufficient to lead to a broad reallocation of sensory resources is unknown but plausible. Several studies have reported rapid changes in visual processing and early visual cortex activity following short-term (i.e., minutes to a few hours) periods of sensory deprivation (*Binda et al., 2018*; *Jamal and Dilks, 2020*; *Keck et al., 2011*; *Kwon et al., 2009*). Gradual improvements in vision with time are usually observed during the first months following refractive (*Kohnen et al., 2004*; *Pesudovs, 2005*) and cataract (*Montés-Micó and Alió, 2003*) surgery, or following correction of relatively low magnitudes of astigmatism (*Vinas et al., 2012*). Moreover, exposure to visual inputs deprived of vertical information over four consecutive days showed limits of neuronal adaptation mechanisms (*Haak et al., 2014*), which peaked during the first day but then dropped in strength despite the adapting environment remaining constant. Such limits of neural adaptation to altered sensory inputs over time are consistent with the notion that sensory processing must be plastic, to swiftly respond to changes in the sensory inputs, but remain stable, so that functions endure.

Another important factor to consider is the fact that participants were tested only in one eye. The two eyes of an individual routinely differ in their optical and neural properties, particularly in disease. However, differences in optical quality between the two eyes of our KC patients did not seem to affect the pattern of results we observed for the tested eye. For instance, the four most severe KC patients (KC7–10) all showed pronounced alterations of visual processing under AO correction, consistent with the large amounts of habitual optical aberrations present in their tested eye. Yet, KC9–10 had well-corrected optical quality in their untested eye, whereas KC7–8 had similar optical deficits in both eyes. Thus, the changes in monocular contrast sensitivity and VA we observed under AO correction were primarily driven by the habitual optical quality of the tested eye. It is likely that an accurate assessment of the optical and neural properties in both eyes would have revealed interocular factors influencing neural compensation to optical blur. Interocular differences between the eyes impact binocular visual functions, reducing binocular summation (*Jiménez et al., 2006*) and stereopsis (*Lam et al., 1996*; *Ng et al., 2021*). Moreover, in individuals with typical optics, perceived focus through either eye is more similar than predicted by the properties of each eye, suggesting the existence of cyclopean neural calibration mechanisms that help reduce perceptual differences between the eyes (*Radhakrishnan et al., 2015*).

Our methods are largely agnostic about the neural bases of long-term exposure to the eye's optics. Physiologically, differences in signal enhancement mechanisms in the signal processing domain (i.e., PTM) could reflect changes in contrast gain of SF-selective neurons (*Dosher and Lu, 2020*). We hypothesize that our findings are likely to be due, at least partially, to neural changes occurring in V1. Pooled responses of V1 neurons can predict behavioral performance in contrast sensitivity tasks (*Boynton et al., 1999*), and perceptual CSF and neuronal CSF in V1 are highly correlated (*Meng et al., 2013*; *Niemeyer and Paradiso, 2017*), sharing a similar shape, amplitude, and peak SF. That is, the shape of the perceptual CSF is assumed to reflect properties of V1 neurons selective to a wide SF range. While responses in the lateral geniculate nucleus are primarily lowpass

in SF, the majority of V1 neurons possess bandpass characteristics. The fall-off of the CSF at low SFs is generally attributed to lateral inhibition in retinal ganglion cells (*Kelly, 1975*), as well as normalization – a canonical neural computation in which the response of an individual neuron is divided by the summed activity of a pool (population) of neurons (*Carandini and Heeger, 2012*). Divisive normalization plays an important role in determining visual sensitivity, serving as a form of homeostatic control of V1 functional properties that optimize the network nonlinearities to the statistical structure of the visual input (*Fournier et al., 2011*). Here, severe optical blurring in KC eyes would reduce the contribution of high-SF neurons to the normalization pool, in turn increasing the response of neurons selective to lower SFs. Another, not mutually exclusive, neural account could involve interactions between SF-selective channels (*Bauman and Bonds, 1991*; *Bredfeldt and Ringach, 2002*; *De Valois and Tootell, 1983*; *Webster, 1999*). SF-specific inhibitory mechanisms refine SF selectivity of V1 neurons (*Bauman and Bonds, 1991*). For instance, SF-specific inhibition of low SFs plays a key role in the generation of bandpass V1 selectivity (*Bredfeldt and Ringach, 2002*; *De Valois and Tootell, 1983*), shifting preferred spatial selectivity to higher SFs. Moreover, adaptation effects at different spatial scales are not independent (*Webster, 1999*), ruling out simple linear filter models of cortical processing. Changes in normalization and/or SF-specific inhibition mechanisms would involve changes in neural sensitivity across many V1 neurons selective to a wide range of SFs, consistent with the SF-specific changes in sensitivity we observed in KC patients. Note that we cannot unequivocally rule out other types of plasticity that could affect the shape of the CSF, such as changes in the number of neurons selective for different SFs (*DeValois and DeValois, 1988*).

Finally, the effects of prolonged exposure to poor optical quality likely reflect the interaction of multiple mechanisms over time, from short-term adaptation effects to long-term learning experiences. Following short-term blur adaptation, humans with typical optical quality usually show enhanced high-SF sensitivity and reduced low-SF sensitivity (*Rajeev and Metha, 2010*; *Webster, 1999*). This pattern of results suggests that the visual system adapts to preserve functional homeostasis at first, adjusting contrast gain to better extract weak high-SF signals in the presence of blur. However, our results show that blur adaptation over considerably longer time periods has opposite effects on contrast sensitivity. Although distinct neural adaptation mechanisms have been shown to operate over different timescales (*Bao and Engel, 2012*; *Bao et al., 2013*), the effects of adaptation usually weaken over a span of days but remain qualitatively the same (*Haak et al., 2014*). One possible explanation is that neural compensation following long-term exposure to altered sensory inputs relies on the interactions between sensory adaptation and perceptual learning. Under natural viewing, KC patients have to learn how to best use the severely degraded retinal inputs to interact with their environment, which is extremely rich in terms of tasks and stimulus properties. Visual adaptation (*Webster, 2011*; *Webster, 2015*) and perceptual learning (*Dosher and Lu, 2017*; *Kumano and Uka, 2013*) are two forms of experience-dependent plasticity that can interact, despite having distinct neural mechanisms and perceptual consequences. For instance, training over multiple days has been shown to weaken the perceptual effects of visual adaptation (*Dong et al., 2016*). Moreover, perceptual learning is able to reconfigure the effects of visual adaptation (*McGovern et al., 2012*). While adaptation reduced sensitivity before training, learning while in an adapted state reversed the effects of adaptation, leading to an overall benefit following training (*McGovern et al., 2012*). This reversal in adaptation effects following repetitive training was specific to the trained adapted state, while untrained adapted states were associated with significant costs in sensitivity following training (*McGovern et al., 2012*). Such interactions between sensory adaptation and perceptual learning mechanisms may account for the qualitative differences between the effects of short-term and long-term exposure to blur on contrast sensitivity.

The atypical CSF and poorer acuity observed in KC patients tested under fully-corrected optical conditions reflect neural compensation mechanisms that improve visual processing of the degraded retinal inputs but limit the clinical benefits of improved optical correction. Our findings identified specific mechanisms at play that should be considered in the clinical treatment of patients with optical defects. The eye's optics have been shown to limit the benefits of visual training on contrast sensitivity and VA in adult participants with healthy eyes (*Zhou et al., 2012*). Readaptation under improved optical correction devices could therefore be combined with targeted perceptual learning paradigms (*Sabesan et al., 2017*) to enhance the speed, efficiency, and generality of neural rehabilitation in patients with altered neural processing due to chronic exposure to poor optical quality.

In summary, the present study furthers our understanding of the impact of long-term (i.e., months to years) exposure to severely degraded habitual optics on visual sensitivity to a wide range of SFs. Our findings support the presence of marked neural plasticity in the adult visual system, which allows the brain to longitudinally compensate for optically-related sensory loss. Using AO correction, we were able to bypass optical factors in KC patients and uncovered evidence of a broad, functional reallocation of sensory processing resources mediated by SF-specific changes in signal enhancement mechanisms (i.e., internal additive noise levels). Overall, our results reveal the existence of neural compensation mechanisms that optimize visual processing to the altered retinal inputs, favoring perceptual information least affected by the eye's optics. These findings have clinical implications, showing that optical correction alone is unlikely to fully improve patients' perceptual quality. Instead, clinical rehabilitation approaches should take into consideration changes in visual sensitivity resulting from chronic exposure to poor optics. An important follow-up question in this context will be to assess how fast and to what extent the visual system of KC-afflicted individuals can readapt to perfectly corrected retinal images.

## Materials and methods

### Participants

A total of 10 keratoconus patients (KC1–9; mean age 34.7 ± 12.4, range 22–55) and 14 age-matched control observers with healthy eyes (CTRL1–14; mean age 33.6 ± 12.3, range 21–57) participated in this study (see *Table 1* for demographic information). Six of the KC patients (mean age 35.8 ± 14.5, range 24–55) and eight of the age-matched control observers (mean age 37.5 ± 12.7, range 21–57) participated in experiment 3. Participants were cyclopleged with tropicamide (1%) to dilate the pupil and paralyze accommodation during visual testing. All participants were screened prior to the study by one of our ophthalmologists, providing standard information (e.g., corneal curvature, refractive error) and ensuring that dilation was safe. An additional screening session was performed to ensure we could obtain good quality wavefront measurements and reach stable aberration-free condition under AO correction. Three potential KC patients and one control observer did not pass this second screening stage and were not tested further along. All control observers were emmetropes or low myopes/hyperopes, with well-corrected optical quality in both eyes and no specific issues that may have affected their optical quality in their daily life (e.g., uncorrected or high myopia, dry eye). All KC patients included in the present study had been diagnosed with KC at least 12 months before testing, with a reported onset of the disease around their twenties. Importantly, all patients had been wearing the same habitual corrective method and did not report substantial changes in their vision for at least 12 months before testing. The eye of KC patients with the most severe habitual optical aberrations was selected for monocular testing under AO (except for KC2 due to the presence of scaring in the untested eye). A qualitative description of the optical quality of the untested eye based on the screening performed by the ophthalmologist is provided in *Table 1*. The Research Subjects Review Board at the University of Rochester Medical Center approved all experimental protocols, which were conducted according to the guidelines of the Declaration of Helsinki. Informed written consent was obtained from all subjects prior to participation. Participants were compensated $12/hr.

### Adaptive optics vision simulator

An adaptive optics vision simulator (AOVS) allowed us to bypass any optical factors during psychophysical testing by measuring and correcting all monochromatic and polychromatic aberrations. As illustrated in a simplified schematic (*Figure 2A*; see also *Figure 2—figure supplement 1*), the AOVS consisted of a custom-built Shack–Hartmann wavefront sensor to measure the wavefront aberrations, a deformable mirror (ALPAO DM97, Montbonnot, France) to correct subjects' wavefront aberrations, an artificial pupil set to 5.8 mm, and a calibrated visual display for psychophysical measurements. Wavefront aberrations were measured for a 6-mm pupil at fovea using an infrared (840 ± 20 nm) superluminescent diode. The AOVS was operated in continuous closed loop (~8 Hz), allowing to fully correct all aberrations over a 6-mm pupil using the deformable mirror. Visual testing was performed at fovea under white light conditions using a modified digital light processor display (Sharp XR-10X, Abeno-ku, Osaka, Japan) operating at 8 bits with 1024 × 768 (60 Hz) resolution, sustending 3.56 ×

2.67 degrees of visual angle (dva). The display was calibrated with a PR-650 SpectraScan Colorimeter (Photo Research, Chatsworth, CA), and luminance precision was increased to 10.7 bits using the bitstealing technique. A dental impression bite bar mounted to motorized translation stages (x, y, z) with adjustable lateral headrests was used to stabilize head position and maintain pupil alignment during visual testing (*Figure 2—figure supplement 1*).

## Optical quality estimation

Wavefront measurements were collected for each observer using their everyday correction method, if any, to estimate each participant's habitual optical quality (*Figure 2B*). Wavefront aberrations were fitted to individual Zernike polynomials up to the 10th order, with 65 Zernike coefficients. The square root of the sum of Zernike coefficients was used to estimate the overall RMS error (total RMS: all Zernike coefficients) and the higher-order aberration (HOAs) RMS error (hRMS+: 6–65th Zernike coefficients), reported in microns (μm). Wavefront measurements were also collected during visual testing to estimate the quality and stability of the AO correction (*Figure 2C*). The MTF was computed for each participant under their habitual optical quality (*Figure 2D*) and under full AO correction (*Figure 2E*). Specifically, we computed the optical transfer function (OTF) from the individual subject's aberrations at a center wavelength of 555 nm and for a pupil diameter of 6 mm. The MTF was calculated as the absolute value of the OTF, which was averaged radially. Full correction of all Zernike coefficients was applied for the entire duration of each testing session, with the exception of defocus for which appropriate defocus values were used to correct axial chromatic aberrations. Subjective defocus values were determined for all participants by asking them to adjust defocus using a motorized Badal prism to make a high-contrast Snellen letter 'E' as sharp as possible, while correcting all other aberrations using AO. Then, through-focus high-contrast VA measurements were used to objectively verify that the subjective defocus value applied during AO correction resulted in a very focused, perceived image and maximal VA performance. After this initial session, each experimental session started with VA measurements to ensure the quality and stability of the AO correction before data collection. To maximize AO correction during stimulus presentation, participants were trained to blinks between trials and to stop if the perceptual quality got unstable and/or poor quality. To minimize the influence of blinks when estimating the residual RMS under AO correction, the median RMS was computed across time for each measurement (*Figure 2—figure supplement 2* and *Figure 2—video 1*). Then, the average residual RMS was computed for each participant from multiple wavefront measurements collected under AO correction (*Figure 2C*). As detailed and demonstrated in previous work (e.g., *Ghosh et al., 2017*; *Ng et al., 2021*; *Sabesan et al., 2017*; *Sabesan and Yoon, 2009*; *Sabesan et al., 2012*; *Zheleznyak et al., 2016*), our AOVS provided stable, aberration-free optical quality during visual testing in both typical and severely aberrated eyes. Statistical analyses indicated a violation of normality for the habitual total RMS of control and KC groups, as well as for the AO-corrected residual RMS of KC observers (Shapiro–Wilk test, p<0.05). A Mann–Whitney U test was therefore used to compare control and KC groups for these variables, with the effect size given by the rank biserial correlation. Welch's *t*-tests were used to compare the amount of habitual hRMS+ and vertical coma between groups due to unequal variance between groups (Levene's test, p<0.05), with the effect size given by Cohen's *d*.

## Experiment 1: contrast sensitivity measurements

The CSF of each participant was measured using a 2-AFC orientation discrimination task (*Figure 3A*). Each trial began with a dynamic fixation point. After a blank screen, a Gabor stimulus (Gaussian envelope SD 0.75 dva; SF range 0.25–30 cpd with 12 equal log-step values) oriented ±45˚ was presented at fovea. A 500-ms temporal Gaussian envelope was used to blend stimuli into the background and avoid onset/offset transients, and a brief tone signaled stimulus onset to reduce temporal uncertainty. Participants were asked to report whether the stimulus was tilted clockwise or counterclockwise. Auditory feedback was provided for both correct and incorrect responses. We used the *qCSF* method (*Hou et al., 2010*; *Hou et al., 2016*; *Lesmes et al., 2010*) to estimate 81%-correct contrast thresholds over a broad SF range. The *qCSF* method is a Bayesian adaptive strategy using a priori knowledge about the CSF's general form to obtain reasonably accurate estimates of sensitivity across SFs with as little as 100 trials. The *qCSF* method describes the CSF as a *truncated log-parabola* with four parameters (*Figure 3B*): (1) the peak sensitivity (amplitude) $CS_{max}$, (2) the

peak frequency $SF_{\text{peak}}$, (3) the bandwidth $\beta$ (full width in octaves at half maximum), and (4) the truncation level (plateau) at low SF $\delta$. Without truncation, the CSF is defined as a function of the stimulus frequency ($f$) in decimal log as a *log-parabola CS'(f)*:

$$CS'(f) = \log_{10}(CS_{\text{max}}) - \log_{10}(2)\left(\frac{\log_{10}(f) - \log_{10}(SF_{\text{peak}})}{\log_{10}(2\beta)/2}\right)^2 \tag{1}$$

This *log-parabola* is then truncated at SF below the peak SF with the truncation parameter $\delta$, which determines contrast sensitivity at low SF ($CS_{\text{low}}$):

$$\begin{aligned} CS(f) &= CS'(f), & f \geq SF_{\text{peak}}, \\ CS_{\text{low}}(f) &= \log_{10}(CS_{\text{max}}) - \delta, & f < SF_{\text{peak}} \;\; \text{and} \;\; CS'(f) < CS_{\text{max}} - \delta \end{aligned} \tag{2}$$

In addition, the high-SF cutoff ($SF_{\text{cutoff}}$) was estimated from the qCSF fits, corresponding to the frequency at which $CS(SF_{\text{cutoff}}) = 0$ (i.e., 100% contrast). The test procedure was similar to that described in previous studies (e.g., *Hou et al., 2010*; *Lesmes et al., 2010*). Briefly, the stimulus space consisted of gratings with contrasts ranging from 0.1% to 99% in steps of 1.5 dB and SF from 0.25 to 30 cpd. After familiarization with the task, participants performed around 7 qCSF runs of 100 trials each (mean number of runs: 7±4 for CTRLs, 7±3 for KCs), which were then combined to compute individual qCSF functions using all trials. Confidence intervals (95% CIs) and p-values were computed from bootstrapping. Specifically, individual trials were randomly resampled with replacement to generate a resampled trial sequence, which was refitted using the qCSF procedure. We repeated this procedure of resampling and refitting 10,000 times to generate bootstrap distributions of the fitted parameters, along with associated confidence intervals. To assess statistical significance for differences in qCSF parameters and SF sensitivity estimates between groups, we computed the difference from the 10,000 random pairs of values from the bootstrap distributions of the two groups, and defined p-values as the proportion of samples that 'crossed' zero. Note that for all of the key comparisons significant p-values also correspond to non-overlapping confidence intervals. For group comparisons at each of the 12 SF levels, p-values were corrected for multiple comparisons using Bonferroni correction by dividing the alpha value level (*p < 0.05; **p < 0.01; ***p < 0.001) by the number of SF levels (i.e., 12).

## Experiment 2: VA measurements

VA thresholds were measured using a 4-AFC letter orientation task (*Figure 5*), in which observers judged whether a high-contrast Snellen 'E' letter presented for 250 ms at fixation was oriented upward, downward, leftward, or rightward. Stimuli were black letters presented on a white background. The size of each Snellen E letter was adjusted from trial to trial using the QUEST staircase method (*Watson and Pelli, 1983*) to estimate 62.5%-correct VA thresholds (in logMAR), with 40 trials per staircase. Multiple VA thresholds were collected under AO correction for each participant (mean number of runs: 7 ± 4 for CTRLs, 12 ± 7 for KCs). Each experimental session also started with VA testing to ensure the quality and stability of the AO correction before starting data collection.

## Experiment 3: equivalent noise paradigm

We used an equivalent noise paradigm (*Bejjanki et al., 2014*; *Burgess et al., 1981*; *Dosher and Lu, 1998*; *Lu and Dosher, 2008*; *Park et al., 2017*) where perceptual thresholds are measured as a function of varying external noise levels added to the stimuli. Contrast thresholds were measured using a 2-AFC orientation discrimination task for various amounts of external noise (*Figure 6A*). Each trial began with a dynamic fixation point. After a blank screen, a ±45° oriented Gabor signal (cosine envelope diameter 2 dva) embedded in different intensity levels of dynamic white noise (eight levels, from 0 to 0.33 SD; check size 0.06°) was presented at fovea for 100 ms (six noise frames). Participants were asked to report the orientation (±45° from the vertical) of the Gabor patch presented on each trial. Each stimulus was accompanied with a brief tone to reduce temporal uncertainty, and auditory feedback was provided for both correct and incorrect responses. Stimulus presentation was controlled using the *FAST* method (*Vul et al., 2010*), an advanced adaptive psychophysical technique used to accurately estimate relevant model parameters in just 480 trials per participant for each SF. This allowed us to estimate contrast thresholds as a function of external noise contrast levels for

both 70.71%-correct and 79.37%-correct difficulty levels, similar to previous studies (**Bejjanki et al., 2014**; **Dosher and Lu, 1998**; **Lu and Dosher, 2008**; **Park et al., 2017**). Five different SFs (0.5, 1, 3, 9, and 16 cpd) were tested across different experimental sessions of 480 trials each (divided into four blocks of 120 trials), for a total of 2400 trials per observer. Data from the equivalent noise experiment were pooled from the *FAST* structures to estimate psychophysical thresholds for each participant at each of the external noise levels (60 trials per level). This approach yielded independent, albeit noisy, contrast threshold estimates at each noise level. Thresholds were estimated by fitting a Weibull function at each external noise level independently:

$$P(c) = 1 - (1 - 0.5) * 2^{-\left(\frac{\log(c)}{\alpha}\right)^{\eta}}$$

(3)

where $P$ denotes percent correct, $c$ is stimulus contrast, $\alpha$ is contrast threshold at 75%-correct performance level, and $\eta$ is the slope of the function. Similar to Park and colleagues (2017), we used a Bayesian model fitting method implementing a Markov Chain Monte Carlo (MCMC) technique to estimate the two free parameters of the Weibull function ($\eta$ and $\alpha$). Specifically, we sampled the posterior distributions of the parameters using JAGS software (http://mcmc-jags.sourceforge.net). We assumed a broad uniform prior on each parameter with a range that includes all practically possible values. Maximum a posteriori (the mode of the posterior) were used as the best estimates of the model parameters. We discarded the first 15,000 samples as a burn-in period and thinned the samples to reduce correlations by only selecting every 200 samples. A total of 10 chains were run in parallel, resulting in 1000 posterior samples per chain. Thresholds at 70.71% and 79.37% were then computed from the estimated Weibull functions.

## Experiment 3: conventional PTM analysis

The PTM model (**Dosher and Lu, 1998**; **Lu and Dosher, 2008**) was used to estimate the sources of signal-to-noise changes responsible for performance differences between KC and age-matched control observers. Contrast thresholds for two difficulty levels (70.71% and 79.37%) and across external noise levels were fitted with the PTM to quantify the effects of noise on contrast perception. The PTM consists of five main components (*Figure 6B*): a perceptual template tuned to the signal, a nonlinear transducer function, a multiplicative internal noise source ($N_{mul}$), an additive internal noise source ($N_{add}$), and a decision process. The output of the perceptual template is processed by two pathways: the signal pathway in which the output is processed by an expansive nonlinear transducer function, and the multiplicative internal noise pathway in which the output is processed by a rectified nonlinear transducer function. Multiplicative noise is an independent noise source whose amplitude is proportional to the (average) amplitude of the output from the perceptual template, acting as a gain control mechanism. Additive internal noise is another noise source whose amplitude does not vary with signal strength and is related to the gain of the perceptual template. Both multiplicative and additive noise sources are added to the output from template matching, and the noisy signal is submitted to a decision process. Here, contrast thresholds ($C_\tau$) are characterized by

$$C_\tau = \frac{1}{\beta} \left[ \frac{\left(1 + N_{mul}^2\right) N_{ext}^{2\gamma} + N_{add}^2}{\left(1/d'^2 - N_{mul}^2\right)} \right]^{\frac{1}{2\gamma}}$$

(4)

where the input (signal + external noise $N_{ext}$) is filtered through a perceptual template, resulting in the amplification of the signal via a gain factor $\beta$. The output of the filter is then transformed through a nonlinear transducer function that amplifies the inputs to the $\gamma^{th}$ power, with both internal additive noise ($N_{add}$) and multiplicative internal noise ($N_{mul}$) being added to the output. $N_{add}$ remains constant across signal levels, while $N_{mul}$ is proportional to the signal strength. $N_{ext}$ is manipulated by the experimenter along with the input signal. Finally, a decision process determines the contrast threshold at a specific performance level ($d'$). The PTM (**Dosher and Lu, 1998**; **Dosher and Lu, 2000**; **Lu and Dosher, 2008**) considers that performance differences result from changes in three possible sources of inefficiency (*Figure 6B, C*): (1) changes in internal additive noise ($A_{add}$), equivalent to changes in signal enhancement (amplification) of the perceptual template; (2) changes in external noise filtering ($A_{ext}$), corresponding to changes in signal selectivity (tuning) of the perceptual template; and (3) changes in multiplicative internal noise ($A_{mul}$), acting as a gain control mechanism

compressing the perceptual template's responses to signal contrast (nonlinearity change). Group averages of the independently estimated thresholds were used to fit the conventional PTM across SFs. To characterize group differences between CTRL and KC observers at each SF, the PTM introduces three coefficient indices ($A_{mul}$(SF,group), $A_{add}$(SF,group), and $A_{ext}$(SF,group)) to *Equation (4)*:

$$C_\tau = \frac{1}{\beta}\left[\frac{\left(1+(A_m(SF,group)*N_{mul}(SF))^2\right)\left(A_e(SF,group)*N_{ext}\right)^{2\gamma}+(A_a(SF,group)*N_{add}(SF))^2}{\left(1/d'^2-(A_m(SF,group)*N_{mul}(SF))^2\right)}\right]^{\frac{1}{2\gamma}} \quad (5)$$

The coefficient indices for CTRL are fixed to 1 ($A_{mul}$(CTRL) = $A_{add}$(CTRL) = $A_{ext}$(CTRL) = 1). $N_{add}$(SF) and $N_{mul}$(SF) did not vary across groups, only across SFs to reflect differences in contrast thresholds related to stimulus SF. To estimate whether and how these three types of noise could account for group differences in KC group(s), we computed the relative differences in the effects of each source of noise between the groups. Elevated sources of noise in KC relative to CTRL would correspond to $A_{mul}$(KC), $A_{add}$(KC), and/or $A_{ext}$(KC) estimates being higher than 1 (i.e., CTRL group), and to poorer contrast thresholds. Conversely, reduced sources of noise in KC relative to CTRL would correspond to estimates lower than 1, and to better contrast thresholds. These three coefficient indices for each KC group could be restricted, either varying or fixed at 1. We compared the results obtained by fitting the eight possible variants of the PTM (*Figure 7*, *Figure 7—figure supplements 1–4*), ranging from no change in any of the three noise reduction mechanisms to the full model with changes in all three noise-reduction mechanisms. We compared the control group to two groups of KC participants (mild/moderate and severe) at five different SFs. The full model consisted of 12 shared parameters across groups (β, γ, $N_{mul}$(SF1-5), $N_{add}$(SF1-5)), and 15 noise parameters ($A_{mul}$(SF1-5), $A_{add}$(SF1-5), $A_{ext}$(SF1-5)) for each of the two KC groups, for a total of 42 parameters. Model fitting was carried out using a least-square method, and goodness of fit ($r^2$) for each candidate model was computed as follows:

$$r^2 = 1 - \frac{\sum\left[\log\left(C_\tau^{predicted}\right)-\log(C_\tau)\right]^2}{\sum\left[\log(C_\tau)-mean(\log(C_\tau))\right]^2} \quad (6)$$

where $\sum$ and *mean* were computed across groups, SFs, external noise levels, and difficulty levels. Given the large differences in the number of free parameters across models, we computed the Bayesian Information Criterion (BIC) as a criterion for model selection. The BIC selects the best model while correcting for overfitting by introducing a penalty term for the number of free parameters in the model:

$$BIC = n*\log\left(\frac{RSS}{n}\right)+k*\log(n) \quad (7)$$

where $n$ is the number of predicted data points, $RSS$ is the residual sum of squares of the model, and $k$ is the number of parameters of the model. BIC approximates a transformation of a model's posterior probability. We then computed ΔBIC as

$$\Delta BIC^k = BIC^k - BIC^{k^*} \quad (8)$$

where $k^*$ corresponds to the model with the lowest BIC value (i.e., the best model).

## Experiment 3: hierarchical Bayesian PTM analysis

To better assess the link between habitual optical quality and individual variability in the PTM noise estimates, we used a hierarchical Bayesian modeling technique to fit the PTM to each participant. This technique assumes that each participant is drawn from a population distribution (CTRL or KC), which increases statistical power. Importantly, this technique allows estimation of PTM parameters for each participant within a population, providing a better understanding of individual variability. We assumed that an individual's response for a given trial is drawn from a Bernoulli distribution (i.e., $response_{ijk} \sim Bernoulli(\theta_{ijk})$), where $i$ is the difficulty level, $j$ is the external noise level ($N_{ext}$), and $k$ is the trial number. $\theta_{ijk}$ corresponds to the probability of making a correct response:

$$\theta_{ijk} = 1 - (1 - 0.5) * e^{\left(\frac{m_i \log\left(x_{ijk}\right)}{\log\left(C_{\tau_{ij}}\right)}\right)^{\eta}} \tag{9}$$

where $x$ corresponds to the stimulus contrast, $\eta$ to the slope, $C_{\tau_{ij}}$ to the contrast threshold, and $m_i$ to

$$m_i = -\log\left(\frac{1 - q_i}{1 - 0.5}\right)^{\frac{1}{\eta}} \tag{10}$$

The parameter $q_i$ corresponds to the difficulty level (i.e., either 70.71% or 79.37%). The PTM defined the contrast threshold $C_{\tau_{ij}}$ as

$$C_{\tau_{ij}} = \frac{1}{\beta}\left[\frac{\left(1 + N_{mul}^2\right)\left(w_{ext} N_{ext_{ij}}\right)^{2\gamma} + N_{add}^2}{\left(1/d_i'^2 - N_{mul}^2\right)}\right]^{\frac{1}{2\gamma}} \tag{11}$$

*Equation (11)* is the same as *Equation (4)*, except for the fact that a coefficient for external noise ($w_{ext}$) is added here to characterize the impact of external noise on individual's contrast thresholds. For each participant, we estimated three parameters $N_{mul}$, $N_{add}$, and $w_{ext}$. We assumed a fixed β (1.25) and γ (2) for all participants to simplify the model, which was within a reasonable range reported in previous studies (e.g., *Dosher and Lu, 1998*; *Lu and Dosher, 2008*; *Park et al., 2017*). Note that similar results were observed using other values of β and γ. Similar to the Weibull fitting procedure, an MCMC technique was used to sample from the posterior distributions and estimate the free parameters. Here, we assumed hierarchical priors on each of the model parameters, meaning that each model parameter that characterized an individual participant was assumed to be drawn from an independent Gaussian population distribution with a mean and SD that characterized each group (CTRL or KC). Priors for population means and SDs were set to broad uniform distributions. Maximum a posteriori were used as the best parameter estimates for both individual and population parameters. We used 15,000 iterations for burn-in and only selected every 200 samples for thinning. Ten chains were run in parallel, each of which sampled 2000 posterior samples.

## Acknowledgements

This work was supported by NIH/NEI grants RO1 EY014999 (GY, DT, KRH, AB), RO1 EY027314 (KRH, DT, AB), P30 EY001319 (core grant to the Center for Visual Science, University of Rochester), Research to Prevent Blindness, and Schmitt Program on Integrative Neuroscience from the University of Rochester Medical Center (postdoctoral fellowship to AB). We thank Dr. Tara C Vaz, OD, and Dr. Len Zheleznyak for their help with participant screening and testing, as well as Olga Pikul for her help with recruitment of study participants.

## Additional information

### Funding

| Funder | Grant reference number | Author |
|---|---|---|
| National Institutes of Health | R01 EY014999 | Geunyoung Yoon<br>Antoine Barbot<br>Duje Tadin<br>Krystel R Huxlin |
| National Institutes of Health | R01 EY027314 | Krystel R Huxlin<br>Duje Tadin<br>Antoine Barbot |
| National Institutes of Health | P30 EY001319 | Geunyoung Yoon<br>Krystel R Huxlin<br>Duje Tadin |
| Research to Prevent Blindness | | Geunyoung Yoon<br>Krystel R Huxlin<br>Duje Tadin |

| Schmitt Program on Integrative Neuroscience | Postdoctoral Fellowship | Antoine Barbot |

The funders had no role in study design, data collection and interpretation, or the decision to submit the work for publication.

## Author contributions

Antoine Barbot, Conceptualization, Software, Formal analysis, Funding acquisition, Validation, Investigation, Visualization, Methodology, Writing - original draft, Project administration, Writing - review and editing; Woon Ju Park, Software, Formal analysis, Visualization, Methodology, Writing - review and editing; Cherlyn J Ng, Investigation, Writing - review and editing; Ru-Yuan Zhang, Software, Formal analysis, Validation, Methodology, Writing - review and editing; Krystel R Huxlin, Conceptualization, Supervision, Funding acquisition, Methodology, Project administration, Writing - review and editing; Duje Tadin, Conceptualization, Software, Formal analysis, Supervision, Funding acquisition, Validation, Visualization, Methodology, Writing - original draft, Project administration, Writing - review and editing; Geunyoung Yoon, Conceptualization, Software, Formal analysis, Supervision, Funding acquisition, Investigation, Visualization, Methodology, Writing - original draft, Project administration, Writing - review and editing

## Author ORCIDs

Antoine Barbot (iD) https://orcid.org/0000-0002-3301-4279
Woon Ju Park (iD) https://orcid.org/0000-0003-1473-4876
Cherlyn J Ng (iD) https://orcid.org/0000-0001-5816-8744
Ru-Yuan Zhang (iD) https://orcid.org/0000-0002-0654-715X
Krystel R Huxlin (iD) https://orcid.org/0000-0001-7138-6156
Duje Tadin (iD) http://orcid.org/0000-0003-1571-5589
Geunyoung Yoon (iD) https://orcid.org/0000-0003-4171-8927

## Ethics

Human subjects: All experimental protocols were conducted according to the guidelines of the Declaration of Helsinki and approved by The Research Subjects Review Board at the University of Rochester Medical Center (#53149). Informed written consent was obtained from all participants prior to participation. Participants were compensated $12/hour.

## Decision letter and Author response

Decision letter https://doi.org/10.7554/eLife.58734.sa1
Author response https://doi.org/10.7554/eLife.58734.sa2

# Additional files

## Supplementary files

• Transparent reporting form

## Data availability

All data are available from the OSF database URL: https://osf.io/p67hy/.

The following dataset was generated:

| Author(s) | Year | Dataset title | Dataset URL | Database and Identifier |
|---|---|---|---|---|
| Barbot A | 2021 | Functional reallocation of sensory processing resources caused by long-term neural adaptation to altered optics | https://osf.io/p67hy/ | OSF, p67hy |

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
