## [Decision Letter]

**Acceptance summary:**

By using advanced optical technique, combined with psychophysical and modelling methods, this study addresses an important question relating to the functionality of neural responses to external stimuli after long-term optical defects. Altered contrast sensitivity functions, visual acuity and internal noisy properties of keratoconus participants all compared with that of matched control observers are reported, providing novel and key information for understanding sensory plasticity.

**Decision letter after peer review:**

Thank you for submitting your article "Functional reorganization of sensory processing caused by long-term neural adaptation to altered optics" for consideration by *eLife*. Your article has been reviewed by three peer reviewers, including Ming Meng as the Reviewing Editor and Reviewer #1, and the evaluation has been overseen by Tamar Makin as the Senior Editor. The following individual involved in review of your submission has agreed to reveal their identity: Jiawei Zhou (Reviewer #3).

The reviewers have discussed the reviews with one another and the Reviewing Editor has drafted this decision to help you prepare a revised submission.

Summary:

The manuscript entitled "Functional reorganization of sensory processing caused by long-term neural adaptation to altered optics" bypassed the constrains of optical defects by using the adaptive optics system to directly access functional changes on spatial frequency (SF) sensitivity in keratoconus (KC) patients. The current manuscript reported that altered contrast sensitivity functions, visual acuity and internal noisy properties of KC participants all compared with that of matched control observers. In general, KC participants showed enhanced sensitivity to low-SF filters and impaired sensitivity to high-SF filters.

Overall, the current article is well written and structured. By using advanced optical technique, the study addresses a key question relating to the functionality of neural responses to external stimuli after long-term optical defects. The study has used multiple approach to investigate this question, including psychophysical and modelling methods. The data was well analysed and reported, and the results in general support the authors' hypotheses. However, the reviewers reached a consensus that several essential revisions listed below are needed before we can recommend publication.

Essential revisions:

1) More age-matched neurotypical observers are recommended to boost external validity. There appears to be large individual difference among KC patients. Therefore, the decision to divide them to mild/moderate/severe KC groups is understandable. However, when divided to mild/moderate/severe KC groups, each group has merely n=3. What could have caused the individual differences? On the one hand, KC severity certainly could be a reason. On the other hand, confounding effects such as age and case history cannot be easily ruled out with n=3. With such a small sample size, it is simply impossible to have sufficient power to meaningfully evaluate between subject factors. It is understandable that testing more KC patients may not be feasible at the present situation. To show that, in addition to age, other between subject-factors would not matter significantly, one possible solution could be testing 2 or 3 age-matched neurotypical observers for each KC patient who had been tested.

2) Discuss possible effects of case history. More details of KC patients and neurotypical observers should be provided. For example, the affected eyes (unilateral or bilateral KC), age of onset, duration of disease, treatment history, fusion, etc. The authors also need to provide the definition of “neurotypical” for the controls. Did the NT subjects have other issues that might produce blur in their daily life (e.g., uncorrected myopia, high myopia)? These details would be critical in this study. For example, if the "long-term exposure" drives the neuronal changes, a correlation between patients' visual performance and the duration of disease would be expected. Given the small sample size, a correlation analysis probably would not reveal much meaningful information. However, analyzing individual case history could potentially provide valuable insights on constraints of the present finding. In addition, despite that adaptation and perceptual learning were discussed, it is not clear how long would be considered by the authors as the "long-term" effect. Presently, the authors could only assume their findings were caused by long-term exposure to degraded optics. However, it is not clear how sensitivity to SFs may be affected by short-term (but perhaps longer and presumably different than typical short-term adaptation and perceptual learning) exposure to degraded optics. A few studies have shown that topographic reorganization of visual cortex can be quite rapid (Dilks, Baker, Liu, & Kanwisher, 2009; Jamal & Dilks, 2020; cf. Keck et al., 2011). Recently, effects of short-term monocular deprivation were also reported for the human adult visual cortex (Binda et al., 2018). These important considerations need to be addressed.

Binda P, Kurzawski JW, Lunghi C, Biagi L, Tosetti M, Morrone MC. Response to short-term deprivation of the human adult visual cortex measured with 7T BOLD. e*Life*. 2018;7:e40014.

Dilks, D.D., Baker, C.I., Liu, Y., & Kanwisher, N. (2009). "Referred Visual Sensations": Rapid Perceptual Elongation after Visual Cortical Deprivation. Journal of Neuroscience, 29, 8960-8964.

Jamal, Y.A. & Dilks, D.D. (2020). Rapid topographic reorganization in adult human primary visual cortex (V1) during noninvasive and reversible deprivation. Proceedings of the National Academy of Sciences.

Keck et al., (2011). Loss of sensory input causes rapid structural changes of inhibitory neurons in adult mouse visual cortex. Neuron 71, 869-882.

3) Clarifying or altering claims to address these issues:

a) The authors describe the effects as "large-scale functional reorganization of sensory resources" but it is not clear what the scale is for "large". The changes in the CSF are not larger than one could generate from short-term adaptation, or perhaps perceptual learning, and whether they reflect a reorganization or simple gain controls is uncertain. Although the link between altered optical and visual plasticity (e.g., adaptation and perceptual learning) was discussed, note that the scale could be a separate issue. Clarifying the claim for "large-scale" is needed.

b) The changes in the CSF are very intriguing and point to systematic changes in threshold sensitivity. However it is worth noting in the paper that these sensitivity changes are often poor predictors of suprathreshold perception. For example the perceived contrast of visible stimuli is largely independent of frequency, and perceived blur is not a simple function of acuity. For readers unfamiliar with this, the current paper gives the impression that once the CSF is characterized everything else in the KC patients is predictable, and it would be worth emphasizing that there may be many additional adjustments that the present study has not explored.

c) As the authors note, the present effects are in the opposite direction to short-term adaptation. Therefore, the suggestion that this supports the idea of adaptation over multiple timescales is confusing. The Bao and Engel work shows similar changes operating over different timescales, and thus not a qualitatively different type of adjustment as observed here.

d) As also noted the effects may be more consistent with perceptual learning. It would strengthen the paper to explore this idea more fully. For example, are the response changes indicated by the equivalent noise model consistent with the changes found for learning? Are the magnitudes similar? And would experiencing complex natural spectra and tasks be expected to show a transfer of training to threshold detection of single gratings? A clarification is needed.

e) The effects are described in terms of noise changes and signal enhancements, but by the latter do the authors mean signal to noise enhancements? Related to point b), noise changes may be less relevant to suprathreshold vision.

f) According to Figure 2—figure supplement 2, it seems that observers' aberrations were corrected dichoptically. Were they also tested dichoptically? Or, only one eye of each individual was studied? Either way, clarification and discussion on the implication are recommended.

4) Since CS was measured in the current study, it would be better if the authors also provide the modulation transfer functions before and after HOAs correction. Furthermore, the authors might calculate the neural transfer functions (i.e., the difference between MTF and CSF) to better illustrate the difference in neural processing between groups.

---

## [Author Response]

Essential revisions:1) More age-matched neurotypical observers are recommended to boost external validity. There appears to be large individual difference among KC patients. Therefore, the decision to divide them to mild/moderate/severe KC groups is understandable. However, when divided to mild/moderate/severe KC groups, each group has merely n=3. What could have caused the individual differences? On the one hand, KC severity certainly could be a reason. On the other hand, confounding effects such as age and case history cannot be easily ruled out with n=3. With such a small sample size, it is simply impossible to have sufficient power to meaningfully evaluate between subject factors. It is understandable that testing more KC patients may not be feasible at the present situation. To show that, in addition to age, other between subject-factors would not matter significantly, one possible solution could be testing 2 or 3 age-matched neurotypical observers for each KC patient who had been tested.

We have added 5 age-matched controls and 1 KC patient to minimize the age differences between groups and boost power. Three of the five additional neurotypical control observers were older (49, 51, and 57 yo), allowing us to better match the age of the oldest KC patients included in our study (54 and 55 yo). We were also able to add a young KC patient (22 yo) with substantial habitual optical aberrations, who showed poorer visual acuity and altered CSF under AO correction, as expected. Importantly after addition of these 6 new participants, our key results remain unchanged.

In addition, we have added analyses that rule out participant’s age as a confounding factor (Figure 4—figure supplement 2, Figure 5—figure supplement 1). We also cited studies looking at the effects of age on visual processing, and discussed the potential impact of factors such as age and differences in case history. We have also improved the description of the group of control observers used in our study and added more information regarding the case history of the KC patients. We have clarified the fact that, under AO correction, control observers and KC patients can be considered as a single group of neurotypically-developed adults who were chronically exposed to varying levels of optical aberrations past their relevant critical period. KC patients and control observers differed primarily based on the amounts of habitual optical aberrations experienced in their daily life, which we were able to bypass using AO correction. The correlation analyses we report (e.g., Figures 4,5,8) are consistent with this idea. Finally, we note that KC patients were divided into subgroups only for illustration purposes, except for the conventional PTM analysis that is based on group comparisons.

2) Discuss possible effects of case history. More details of KC patients and neurotypical observers should be provided. For example, the affected eyes (unilateral or bilateral KC), age of onset, duration of disease, treatment history, fusion, etc. The authors also need to provide the definition of “neurotypical” for the controls. Did the NT subjects have other issues that might produce blur in their daily life (e.g., uncorrected myopia, high myopia)? These details would be critical in this study. For example, if the "long-term exposure" drives the neuronal changes, a correlation between patients' visual performance and the duration of disease would be expected. Given the small sample size, a correlation analysis probably would not reveal much meaningful information. However, analyzing individual case history could potentially provide valuable insights on constraints of the present finding. In addition, despite that adaptation and perceptual learning were discussed, it is not clear how long would be considered by the authors as the "long-term" effect. Presently, the authors could only assume their findings were caused by long-term exposure to degraded optics. However, it is not clear how sensitivity to SFs may be affected by short-term (but perhaps longer and presumably different than typical short-term adaptation and perceptual learning) exposure to degraded optics. A few studies have shown that topographic reorganization of visual cortex can be quite rapid (Dilks, Baker, Liu, & Kanwisher, 2009; Jamal & Dilks, 2020; cf. Keck et al., 2011). Recently, effects of short-term monocular deprivation were also reported for the human adult visual cortex (Binda et al., 2018). These important considerations need to be addressed.

We now provide a better description of the case history of our KC patients and discuss potential confounding factors. We also improved the description of the control observers to clarify that they did not have any other issues affecting optical quality. Any uncorrected blur would show in the habitual optical aberrations already reported. We agree that the term “neurotypical” was unclear. Age-matched adults with healthy eyes are now referred to as controls (CTRL) in the manuscript. In addition, we showed that participants’ age, which in our case was a proxy of exposure duration, can be ruled out as a confounding factor (see response to Comment 1). This finding suggests that most of the changes in visual processing may occur within the first months or year following the changes in optical quality, and that our results are primarily driven by the severity of the habitual optical aberrations.

3) Clarifying or altering claims to address these issues:a) The authors describe the effects as "large-scale functional reorganization of sensory resources" but it is not clear what the scale is for "large". The changes in the CSF are not larger than one could generate from short-term adaptation, or perhaps perceptual learning, and whether they reflect a reorganization or simple gain controls is uncertain. Although the link between altered optical and visual plasticity (e.g., adaptation and perceptual learning) was discussed, note that the scale could be a separate issue. Clarifying the claim for "large-scale" is needed.

We have now clarified these terms in the title and throughout the manuscript. “Large-scale” refers to the broad range of spatial frequencies over which changes in CSF were observed, rather than the magnitude of the changes. This is now clearly stated. Moreover, we agree that the term “reorganization” may be confusing. To avoid this, we now describe our findings as evidence of a “functional reallocation of sensory processing resources over a wide range of spatial frequencies”.

b) The changes in the CSF are very intriguing and point to systematic changes in threshold sensitivity. However it is worth noting in the paper that these sensitivity changes are often poor predictors of suprathreshold perception. For example the perceived contrast of visible stimuli is largely independent of frequency, and perceived blur is not a simple function of acuity. For readers unfamiliar with this, the current paper gives the impression that once the CSF is characterized everything else in the KC patients is predictable, and it would be worth emphasizing that there may be many additional adjustments that the present study has not explored.

We agree and have clarified the fact that changes in threshold sensitivity do not necessarily reflect changes in suprathreshold perception, and we now discuss the impact of the eye’s optics on contrast constancy.

c) As the authors note, the present effects are in the opposite direction to short-term adaptation. Therefore, the suggestion that this supports the idea of adaptation over multiple timescales is confusing. The Bao and Engel work shows similar changes operating over different timescales, and thus not a qualitatively different type of adjustment as observed here.

We have clarified this aspect of our findings. As pointed out in the next comment, neural compensation mechanisms following long-term exposure to optical aberrations may reflect complex interactions between sensory adaptation and perceptual learning mechanisms, which could account for the qualitative differences between the effects of short-term and long-term blur exposure.

d) As also noted the effects may be more consistent with perceptual learning. It would strengthen the paper to explore this idea more fully. For example, are the response changes indicated by the equivalent noise model consistent with the changes found for learning? Are the magnitudes similar? And would experiencing complex natural spectra and tasks be expected to show a transfer of training to threshold detection of single gratings? A clarification is needed.

As now indicated, neural compensation to the eye’s optics likely reflects interactions between neural adaptation and perceptual learning mechanisms. Indeed, the pattern of gain and losses we observed could reflect learning effects during prolonged exposure to optical blur. Regarding the results of the equivalent noise model, the effects of perceptual learning are generally consistent with a combination of internal additive noise reduction and external noise exclusion (i.e., equivalent to channel reweighting). However, both mechanisms can be independently recruited, and depend on various stimulus properties and training regime, making a direct comparison with our results difficult.

e) The effects are described in terms of noise changes and signal enhancements, but by the latter do the authors mean signal to noise enhancements? Related to point b), noise changes may be less relevant to suprathreshold vision.

We have clarified the fact that internal additive noise reduction and signal enhancement are equivalent in the PTM; they affect the signal-to-noise ratio.

f) According to Figure 2—figure supplement 2, it seems that observers' aberrations were corrected dichoptically. Were they also tested dichoptically? Or, only one eye of each individual was studied? Either way, clarification and discussion on the implication are recommended.

We have clarified the fact that only one eye of each participant was tested and added a segment of Discussion about it. We note that qualitative differences in optical quality between the two eyes did not seem to influence the pattern of results we observed. We also edited the corresponding supplement figure to avoid any confusion.

4) Since CS was measured in the current study, it would be better if the authors also provide the modulation transfer functions before and after HOAs correction. Furthermore, the authors might calculate the neural transfer functions (i.e., the difference between MTF and CSF) to better illustrate the difference in neural processing between groups.

As suggested, we now show the modulation transfer functions of all participants under their habitual optical aberrations and under AO-corrected optical quality (Figure 2D,E). As expected, the presence of habitual optical aberrations resulted in severe reductions of signal amplitude in KC eyes. Importantly, AO correction allowed us to fully correct all optical aberrations in all participants and maintain diffraction limited optical quality. There was no difference in MTF between groups under AO correction. Thus, the differences in CSF observed under AO correction directly reflect differences in neural transfer functions. We have clarified this key aspect of our study.